# Distractor effects in decision making are related to the individual's style of integrating choice attributes

Jing Jun Wong[1], Alessandro Bongioanni[2], Matthew FS Rushworth[3], Bolton KH Chau[1,4,5]*

[1]Department of Rehabilitation Sciences, The Hong Kong Polytechnic University, Hung Hom, Hong Kong; [2]Cognitive Neuroimaging Unit, CEA, INSERM, Université Paris-Saclay, NeuroSpin Center, Gif-sur-Yvette, France; [3]Department of Experimental Psychology, University of Oxford, Oxford, United Kingdom; [4]University Research Facility in Behavioral and Systems Neuroscience, The Hong Kong Polytechnic University, Hung Hom, Hong Kong; [5]Mental Health Research Centre, The Hong Kong Polytechnic University, Hung Hom, Hong Kong

*For correspondence:
boltonchau@gmail.com

**Abstract** Humans make irrational decisions in the presence of irrelevant distractor options. There is little consensus on whether decision making is facilitated or impaired by the presence of a highly rewarding distractor, or whether the distractor effect operates at the level of options' component attributes rather than at the level of their overall value. To reconcile different claims, we argue that it is important to consider the diversity of people's styles of decision making and whether choice attributes are combined in an additive or multiplicative way. Employing a multi-laboratory dataset investigating the same experimental paradigm, we demonstrated that people used a mix of both approaches and the extent to which approach was used varied across individuals. Critically, we identified that this variability was correlated with the distractor effect during decision making. Individuals who tended to use a multiplicative approach to compute value, showed a positive distractor effect. In contrast, a negative distractor effect (divisive normalisation) was prominent in individuals tending towards an additive approach. Findings suggest that the distractor effect is related to how value is constructed, which in turn may be influenced by task and subject specificities. This concurs with recent behavioural and neuroscience findings that multiple distractor effects co-exist.

## eLife assessment

This manuscript provides a **valuable** demonstration that distractor effects in multi-attribute decision-making correlate with the form of attribute integration (additive vs. multiplicative). The evidence supporting the conclusions is **convincing**, but there are questions about how to interpret the findings. The manuscript will be interesting to decision-making researchers in neuroscience, psychology, and related fields.

## Introduction

Psychologists, economists, and neuroscientists have been interested in whether and how decision making is influenced by the presence of unchooseable distractor options. Rationally, choices ought to be unaffected by distractors, however, it has been demonstrated repeatedly that this is not the case in human decision making. For example, the presence of a strongly rewarding, yet unchooseable, distractor can either facilitate or impair decision making (*Chau et al., 2014*; *Louie et al., 2013*; *Webb*

*et al., 2020*). Which effect predominates depends on the distractor's interactions with the chooseable options, which, in turn, are a function of their values (*Chau et al., 2020*).

Intriguingly, most investigations have considered the interaction between distractors and chooseable options either at the level of their overall utility or at the level of their component attributes, but not both (*Chau et al., 2014*; *Chau et al., 2020*; *Gluth et al., 2018*). Recently, however, one study has considered both possible levels of option-distractor interactions and argued that the distractor effect operates mainly at the attribute level rather than the overall utility level (*Cao and Tsetsos, 2022*). When the options are comprised of component attributes (for example, one feature might indicate the probability of an outcome if the option is chosen, while the other might indicate the magnitude of the outcome), it is argued that the distractor exerts its effects through interactions with the attributes of the chooseable options. However, as has been argued in other contexts, just because one type of distractor effect is present does not preclude another type from existing (*Chau et al., 2020*; *Kohl et al., 2023*). Each type of distractor effect can dominate depending on the dynamics between the distractor and the chooseable options. Moreover, the fact that people have diverse ways of making decisions is often overlooked. Therefore, not only may the type of distractor effect that predominates vary as a function of the relative position of the options in the decision space, but also as a function of each individual's style of decision making.

Multiple distractor effects have been proposed. At the level of the overall utility of the choice, the divisive normalization model suggests that the presence of a valuable distractor can impair decision accuracy and this is sometimes known as a negative distractor effect (*Kohl et al., 2023*; *Louie et al., 2013*; *Webb et al., 2020*). Conversely, an attractor network model suggests the opposite – a valuable distractor can improve decision accuracy by slowing down decision speed and this is sometimes known as a positive distractor effect (*Chang et al., 2019*; *Chau et al., 2014*; *Chau et al., 2020*; *Kohl et al., 2023*). At the other level, the level of choice attributes, the selective integration model emphasises that individual attributes of the distractor interact with individual attributes of the chooseable options and distort the way they are integrated. Although these models are sometimes discussed as if they were mutually exclusive, it is possible that some, if not all, of them can co-exist. For example, in a two-attribute decision making scenario, each option and distractor falls at different points in a two-dimensional decision space defined by the two component attributes. Whether the distractor facilitates or impairs decision accuracy depends on the exact locations of the chooseable options and distractor (*Dumbalska et al., 2020*). Alternatively, the decision space can be defined by the sum of and differences in the two chooseable options' overall values, and it has been shown that positive and negative distractor effects predominate in different parts of this decision space (*Chau et al., 2020*; *Kohl et al., 2023*). Hence, it is unlikely that the distractor affects decision making in a single, monotonic way. In addition, although most studies focused on analyzing distractor effects at the group level, they often involved a mix of individuals showing positive and negative distractor effects (*Chang et al., 2019*; *Webb et al., 2020*). It is possible that such variability in distractor effects could be related to variability in people's ways of combining attributes during decision making.

Options that we choose in everyday life are often multi-attribute in nature. For example, a job-seeker may consider the salary and the chance of successfully getting the job before submitting the application. An ideal way of combining the two attributes (the salary and success rate) is by calculating their product, that is, employing the Expected Value model or a multiplicative rule (*Neumann and Morgenstern, 2007*). However, it has been argued that, instead of following this ideal rule, people use a simpler additive rule to combine attributes, in which individual attributes of the same option are added together often via a weighted-sum procedure (*Cao and Tsetsos, 2022*; *Farashahi et al., 2019*). As such, it is easy to take a dichotomous view that people either use a multiplicative or additive rule in their decision making. Intriguingly, however, recently it has been shown that, at the level of each individual human or monkey, decision making involves a combination of both rules (*Bongioanni et al., 2021*; *Scholl et al., 2014*). The two computational strategies may rely on distinct neuronal mechanisms, one in parietal cortex estimating the perceptual differences between stimuli, leading to an additive rule, and a more sophisticated one in prefrontal cortex tracking integrated value, leading to a multiplicative rule. It is possible to devise a composite model by having a single parameter (an integration coefficient) to describe, for each individual decision maker, the extent to which decision-making is based on multiplication or addition. In other words, the integration coefficient captures individual differences in people's and animals' degree of attribute combination during decision making

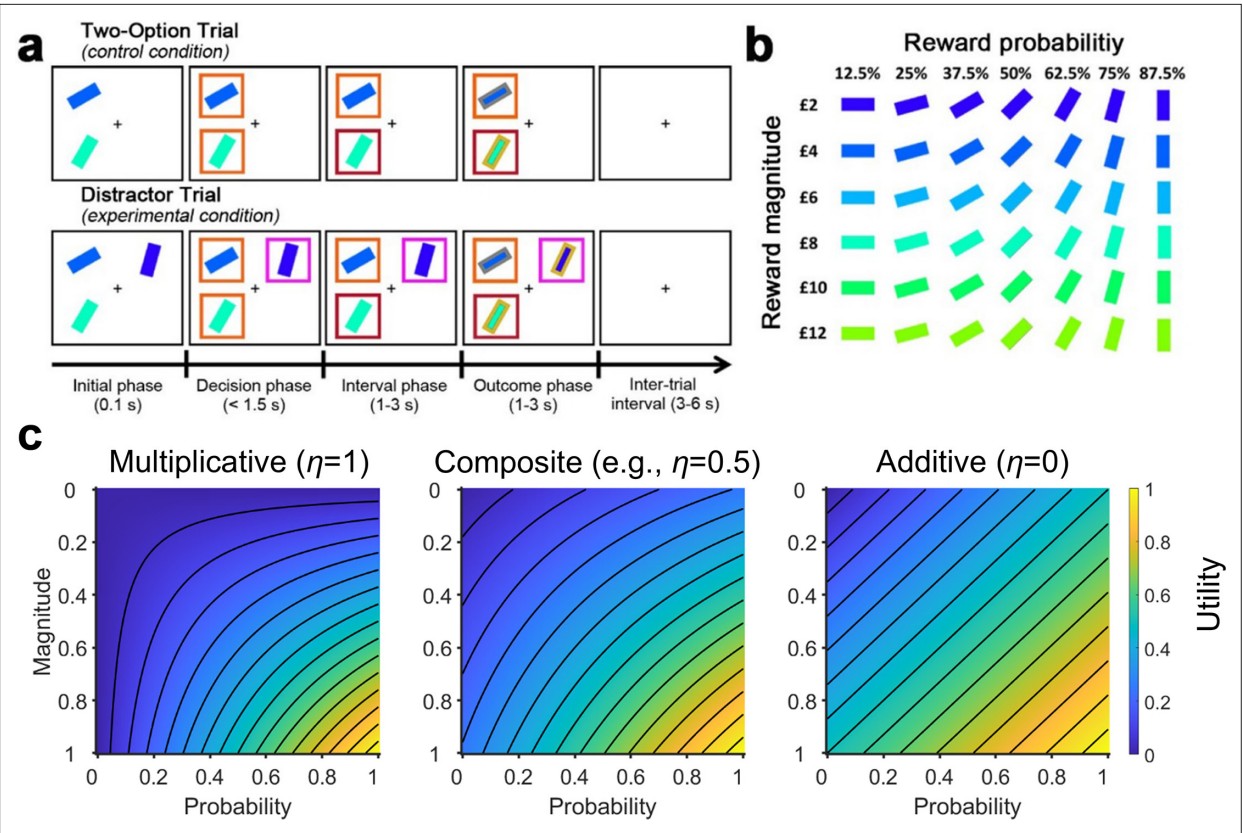

**Figure 1.** The multi-attribute decision making task (*Chau et al., 2014*; *Gluth et al., 2018*). (**a**) On the control Two-Option Trials, participants were presented with two options associated with different levels of reward magnitude and probability, each in the form of a rectangular bar surrounded by an orange box. On the experimental Distractor Trials, three options were presented. Two options were subsequently surrounded by orange boxes indicating that they could be chosen, while a third option was surrounded by a purple box indicating that it was an unchooseable distractor. (**b**) The association between stimulus colour and orientation to reward magnitude and probability, respectively. Participants were instructed to learn these associations prior to task performance. (**c**) Plots illustrating utility estimated using (left) a purely multiplicative rule and (right) a purely additive rule (here assuming equal weights for probability and magnitude). By comparing their corresponding plots, differences in the utility contours are most apparent in the bottom-left and top-right corners. This is because in the multiplicative rule a small value in either the magnitude or probability produces a small overall utility (blue colours). In contrast, in the additive rule the two attributes are independent – a small value in one attribute and a large value in another attribute can produce a moderate overall utility (green colours). (Middle) Here, we included a composite model that combines both rules. The composite model involves an integration coefficient $\eta$ that controls the relative contributions of the multiplicative and additive rules.

(*Figure 1*). For clarity, we stress that the same mathematical formula for additive value can be interpreted as meaning that (1) subjects first estimate the value of each option in an additive way (value integration) and then compare the options, or (2) subjects compare the two magnitudes and separately compare the two probabilities without integrating dimensions into overall values. On the other hand, the mathematical formula for multiplicative value is only compatible with the first interpretation. In this paper we focus on attribute combination styles (multiplicative versus additive) and do not make claims about the order of the operations. More particularly, we consider whether individual differences in combination styles could be related to different forms of distractor effect.

In the current study, we re-analysed data collected from three different laboratories that involved 144 human participants choosing between two options in the presence of a distractor (*Figure 1a and b*; *Chau et al., 2014*; *Chau et al., 2020*; *Gluth et al., 2018*). Recently, the data have been fitted using a multiplicative rule, additive rule, and a multiplicative rule with divisive normalization (*Cao and Tsetsos, 2022*). It was argued that participants' choice behaviour was best described by the additive rule and that the previously reported positive distractor effect was absent when utility was estimated using the additive rule. Here, we fitted the data using the same models and procedures,

but also considered an additional composite model to capture individual variations in the relative use of multiplicative and additive rules (*Figure 1c*). We found that this composite model provides the best account of participants' behaviour. Given the overlap in neuroanatomical bases underlying the different methods of value estimation and the types of distractor effects, we further explored the relationship. Critically, those who employed a more multiplicative style of integrating choice attributes also showed stronger positive distractor effects, whereas those who employed a more additive style showed negative distractor effects. These findings concur with neural data demonstrating that the medial prefrontal cortex (mPFC) computes the overall values of choices in ways that go beyond simply adding their components together, and is the neural site at which positive distractor effects emerge (*Barron et al., 2013*; *Bongioanni et al., 2021*; *Chau et al., 2014*; *Fouragnan et al., 2019*; *Noonan et al., 2017*; *Papageorgiou et al., 2017*), while divisive normalization was previously identified in the posterior parietal cortex (PPC; *Chau et al., 2014*; *Louie et al., 2011*).

## Results

This study analysed empirical data acquired from human participants (five datasets; N=144) performing the multi-attribute decision-making task described in *Figure 1*; *Chau et al., 2014*; *Gluth et al., 2018*. Participants were tasked with maximising their rewards by choosing between options that were defined by different reward magnitudes ($X$) and probabilities ($P$), depicted as rectangular bars of different colours and orientations, respectively (*Figure 1b*). The task involved, as a control, Two-Option Trials, on which participants were offered two options. It also involved Distractor Trials, in which two chooseable options were presented alongside a distractor option that could not be selected. The two chooseable options and the unchosen distractor option are referred to as the higher value ($HV$), lower value ($LV$), and distractor value ($DV$), based on their utility. We began our analyses by assuming utility as Expected Value (i.e. $EV = X \times P$).

### The distractor effect was absent on average

We used a general linear model (GLM), GLM1, to examine whether a distractor effect was present in the choice behaviour of participants. GLM1 included 3 regressors to predict the choice of the HV option: an $HV - LV$ term that represents the choice difficulty (i.e. as the difference in value between HV and LV becomes smaller, it becomes more difficult to select the better option), a $DV - HV$ term that represents the relative distractor value, and a $(HV - LV)(DV - HV)$ interaction term that examines whether the distractor effect was modulated as a function of choice difficulty. Similar approaches have been used previously (*Cao and Tsetsos, 2022*; *Chau et al., 2014*; *Chau et al., 2020*; *Gluth et al., 2018*; *Kohl et al., 2023*). On Distractor Trials (*Figure 2a*), the results of GLM1 showed a positive $DV - HV$ effect [$\beta = 0.0835$, $t(143) = 3.894$, $p = 0.000151$] and $HV - LV$ effect [$\beta = 0.486$, $t(143) = 20.492$, $p < 10^{-43}$], and a negative $(HV - LV)(DV - HV)$ interaction effect

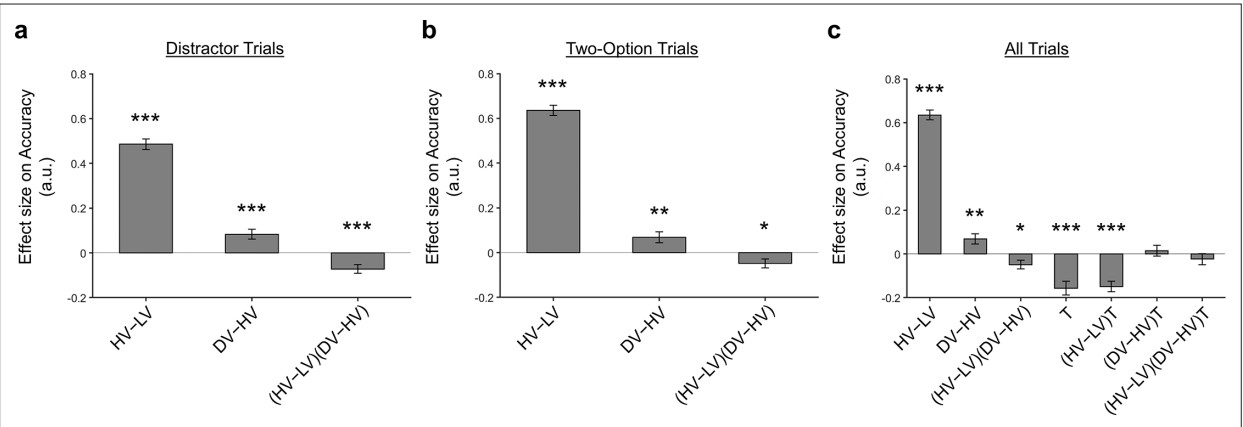

**Figure 2.** A distractor effect was absent, when the way participants combined choice attributes was ignored. Although a positive distractor effect (indicated by the DV-HV term) was present on both (**a**) Distractor and (**b**) Two-Option Trials, (**c**) the distractor effects in the two trial types were not significantly different when they were compared in a single analysis [indicated by the $(DV - HV)T$ term]. Error bars indicate standard error. * p<0.05, ** p<0.01, *** p<0.001.

[$\beta = -0.0720$, $t\left(143\right) = -3.668$, $p = 0.000344$]. However, Cao and Tsetsos (*Cao and Tsetsos, 2022*) suggested that the same analysis should be applied to the control Two-Option Trials. As such, each control trial should be analysed by including a hypothetical distractor identical to the actual distractor that was present in the matched experimental Distractor Trials. Moreover, Cao and Tsetsos suggest specific methods for trial matching (*Cao and Tsetsos, 2022*). Theoretically, a distractor effect should be absent when GLM1 is applied to analyse these control trials because the distractor was not truly present. Consistent with the findings of *Cao and Tsetsos, 2022*, the Two-Option Trials (*Figure 2b*) displayed a significant $DV - HV$ effect [$\beta = 0.0685$, $t\left(143\right) = 2.877$, $p = 0.00463$] and a $\left(HV - LV\right)\left(DV - HV\right)$ interaction effect [$\beta = -0.0487$, $t\left(143\right) = -2.477$, $p = 0.0144$], alongside the expected $HV - LV$ effect [$\beta = 0.635$, $t\left(143\right) = 27.975$, $p < 10^{-59}$].

One key test to examine whether a distractor effect was present was to compare the strength of distractor effects on the experimental Distractor Trials and the control Two-Option Trials. This was achieved by adapting GLM1 into GLM2 and then matching the control Two-Option Trials and Distractor Trials (here we followed exactly the approach suggested by Cao and Tsetsos. More details are presented in the section *GLM analysis of relative choice accuracy*). In addition to the regressors involved in GLM1, GLM2 included a binary variable $T$ to describe the trial type (i.e. 0=Two-Option Trials and 1=Distractor Trial). Then, all original GLM1 regressors were multiplied by this variable $T$. Hence, the presence of a 'stronger' distractor effect on Distractor Trials than on control trials should be manifested in the form of a significant $\left(DV - HV\right)T$ effect or $\left(HV - LV\right)\left(DV - HV\right)T$ effect. However, the results showed that neither the $\left(DV - HV\right)T$ nor the $\left(HV - LV\right)\left(DV - HV\right)T$ term was significant. While these results may seem to suggest that a distractor effect was not present at an overall group level, we argue that the precise way in which a distractor affects decision making is related to how individuals integrate the attributes.

## Identifying individual variabilities in combining choice attributes

During multi-attribute decision making, people integrate information from different attributes. However, the method of integration can be highly variable across individuals and this, in turn, may have an impact on how a distractor effect manifests. Hence, it is imperative to first consider individual differences in how participants integrate the choice attributes. Previous work has demonstrated that choice behaviour may not be adequately described using either additive or multiplicative models, but rather a combination of both (*Bongioanni et al., 2021*). We fitted these models exclusively to the Two-Option Trial data and not the Distractor Trial data, ensuring that the fitting (especially that of the integration coefficient) was independent of any distractor effects, and tested which model best describes participants' choice behaviours. In particular, we included the same set of models suggested by Cao and Tsetsos, which included an EV model, an additive utility (AU) model, and an EV model combined with divisive normalisation. In addition, we included a composite model which utilises an integration coefficient to incorporate both additive and multiplicative methods of combining option attributes. A larger integration coefficient suggests that an individual places more weight on using a multiplicative rule than an additive rule.

When a Bayesian model comparison was performed, the results showed that the composite model provides the best account of participants' choice behaviour (*Figure 3*; exceedance probability = 1.000, estimated model frequency = 0.879). *Figure 3c, d, and e* show the fitted parameters of the composite model: $\eta$, the integration coefficient determining the relative weighting of the additive and multiplicative value ($M = 0.324$, $SE = 0.0214$); $\gamma$, the magnitude/probability weighing ratio ($M = 0.415$, $SE = 0.0243$); and $\vartheta$, the inverse temperature ($M = 9.643$, $SE = 0.400$). Our finding that the average integration coefficient $\eta$ was 0.325 coincides with previous evidence that people were biased towards using an additive, rather than a multiplicative rule. However, it also shows that rather than being fully additive ($\eta$=0) or multiplicative ($\eta$=1), people's choice behaviour is best described as a mixture of both. *Figure 3—figure supplement 1* shows the relationships between all the fitted parameters. *Figure 3—figure supplement 2* reports an additional Bayesian model comparison performed while including a model with nonlinear utility functions based on Prospect Theory (*Kahneman and Tversky, 1979*) with the Prelec formula for probability (*Prelec, 1998*). Consistent with the above finding, the composite model provides the best account of participants' choice behaviour (exceedance probability = 1.000, estimated model frequency = 0.720).

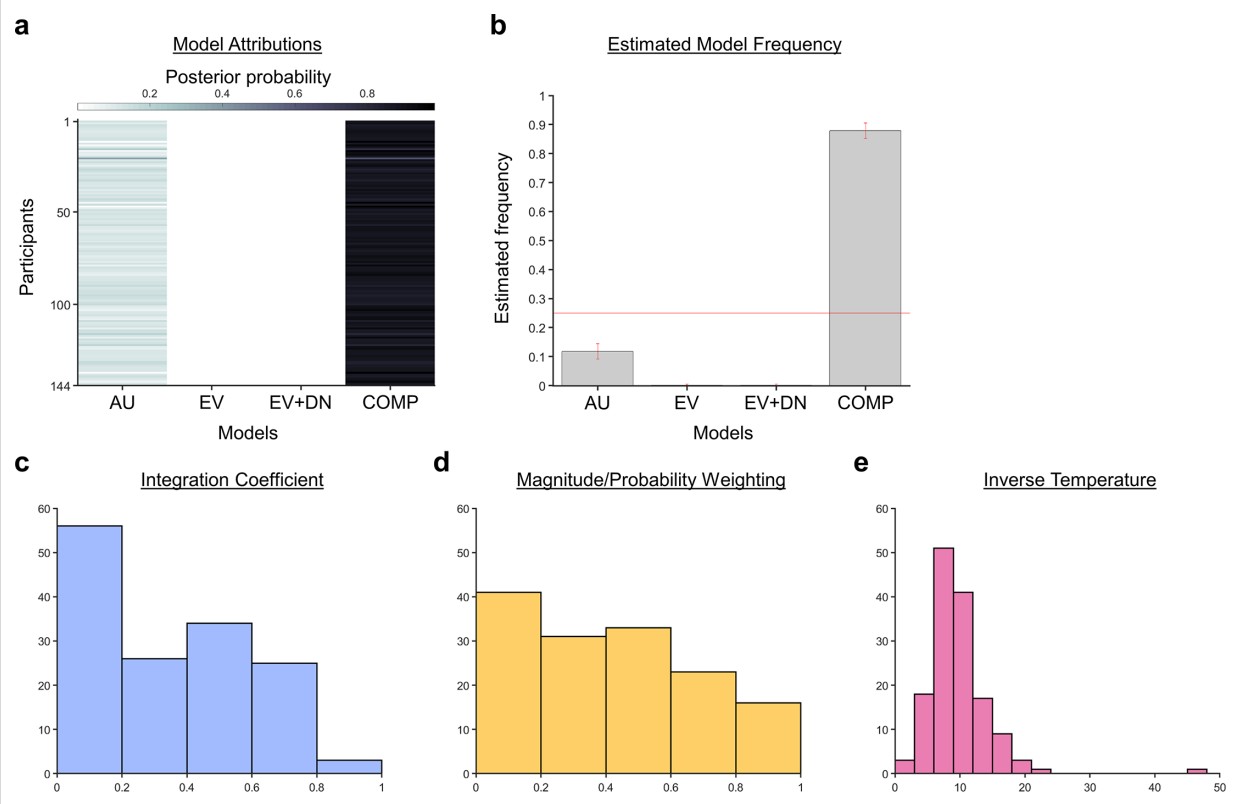

**Figure 3.** A Bayesian model comparison showed that the composite model provides the best account of participants' choice behaviour. (**a**) The posterior probability of each model in accounting for the behaviour of each participant. (**b**) A model comparison demonstrating that the composite model (COMP) is the best fit model, compared to the additive utility (AU), expected value (EV), and expected value and divisive normalisation (EV +DN) models. Histograms showing each participant's fitted parameters using the composite model: (**c**) Values of the integration coefficient ($0 \leq \eta \leq 1$), (**d**) magnitude/probability weighing ratio ($0 \leq \gamma \leq 1$), and (**e**) inverse temperature ($\vartheta$).

The online version of this article includes the following figure supplement(s) for figure 3:

**Figure supplement 1.** Scatterplots illustrating the correlation between the fitted parameters.

**Figure supplement 2.** A Bayesian model comparison showed that the composite model provides the best account of participants' choice behaviour even when a model with utility curvature based on Prospect Theory (*Kahneman and Tversky, 1979*) with the Prelec formula for probability (*Prelec, 1998*) is also considered.

## Multiplicative style of integrating choice attributes was associated with a significant positive distractor effect

It has been shown that evaluations of choices driven by more than just an additive combination of attribute features depend on the medial and/or adjacent ventromedial prefrontal cortex (*Bongioanni et al., 2021*; *Papageorgiou et al., 2017*). On the other hand, a positive distractor effect is also linked to the modulation of decision signals in a similar prefrontal region (*Chau et al., 2014*). On the basis of these findings, we expected that a positive distractor effect may be highly related to the use of a multiplicative method of choice evaluation, given their similar anatomical associations. As such, we proceeded to explore how the distractor effect (i.e. the effect of $(DV - HV)\,T$ obtained from GLM2; *Figure 2c*) was related to the integration coefficient ($\eta$) of the optimal model via a Pearson's correlation (*Figure 4*). As expected, a significant positive correlation was observed [$r\,(142) = 0.282$, $p = 0.000631$]. We noticed that there were 32 participants with integration coefficients that were close to zero (below 1×10⁻⁶). The correlation remained significant even after removing these participants [r(110)=0.202, p=0.0330].

This correlation could be driven by three possible patterns of distractor effect. First, *greater* integration coefficients (i.e. being more multiplicative) could be related to more *positive* distractor effects. Second, *smaller* integration coefficients (i.e. being more additive) could be related to more *negative* distractor effects. Third, both positive and negative distractor effects could be present but

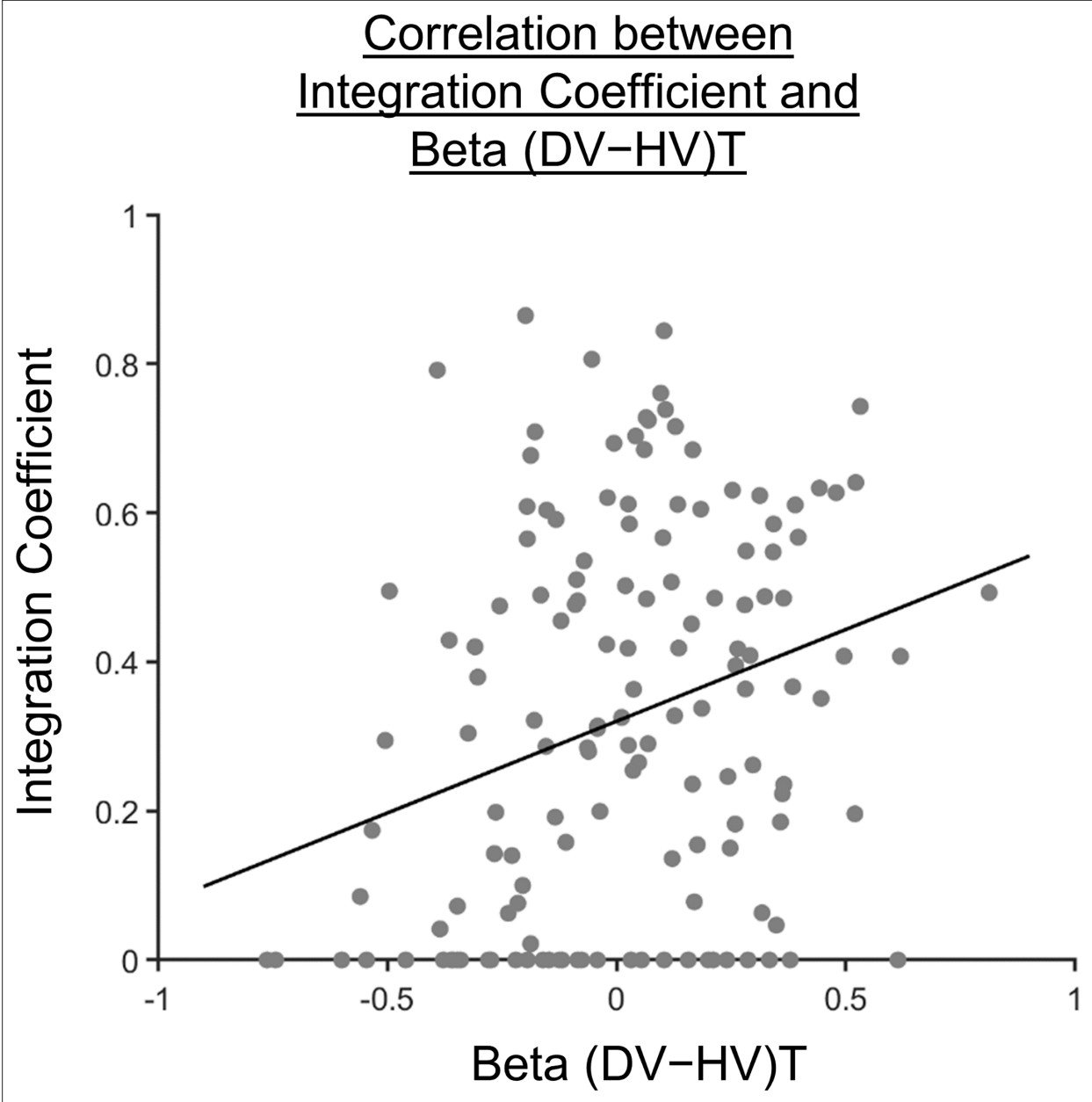

**Figure 4.** Participants showing a more multiplicative style of combining choice attributes into an overall value also showed a greater distractor effect. A scatterplot illustrating a positive correlation between the integration coefficient ($\eta$) and the distractor term (i.e. $(DV - HV)\,T$) obtained from GLM2 (*Figure 2c*). A significant positive correlation was observed [$r\,(142) = 0.282$, $p = 0.000631$].

appeared separately in predominantly multiplicative and additive individuals respectively. To test which was the case, we used the mean integration coefficient value to divide the participants into two groups [Multiplicative Group (N=71) and Additive Group (N=73)]. We then analysed the data from each group using GLM2 (*Figure 5b and c*). We found that the distractor effect, reflected in the $(DV - HV)\,T$ term, was significantly greater in the Multiplicative Group than the Additive Group (*Figure 5*; $t\,(142) = 3.792$, $p = 0.00022$). Critically, the distractor effects were significant within each of the individual groups but bore opposite signs: the distractor effect was positive in the Multiplicative Group [$\beta = 0.105$, $t\,(70) = 3.438$, $p = 0.000991$] but negative in the Additive Group [$\beta = -0.0725$, $t\,(72) = -2.053$, $p = 0.0437$].

Finally, we performed three additional analyses that revealed comparable results to those shown in *Figure 5*. In the first analysis, reported in *Figure 5—figure supplement 1*, we added an $HV + LV$ term

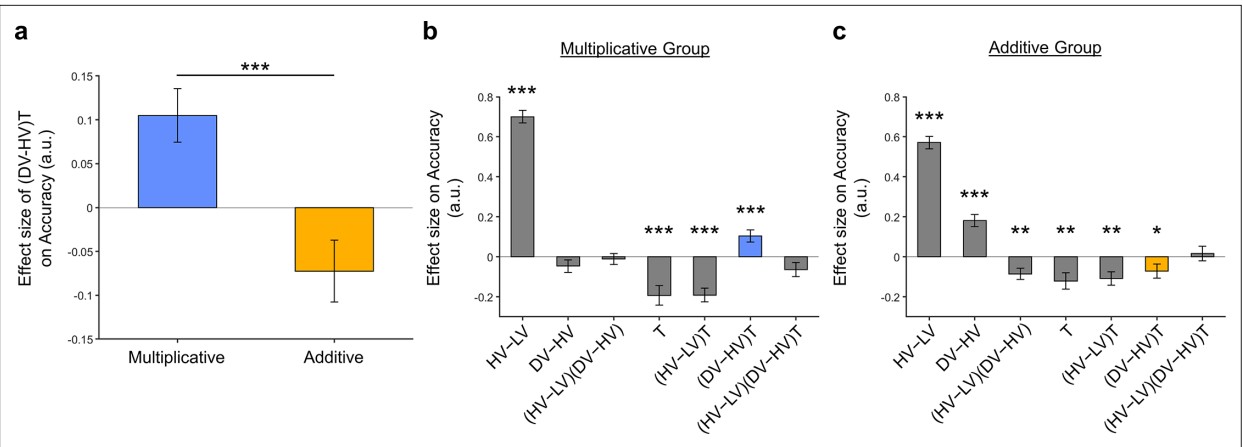

**Figure 5.** Significant distractor effects were found when participants' styles of combining choice attributes were considered. (**a**) A positive distractor effect, indicated by the $(DV - HV)\,T$ term, was found in the Multiplicative Group, whereas a negative distractor effect was found in the Additive Group. The $(DV - HV)\,T$ term was also significantly different between the two groups. Plots showing all regression weights of GLM2 when the data of the (**b**) Multiplicative Group and (**c**) Additive Group were analysed. The key $(DV - HV)\,T$ term extracted in (**a**) is highlighted in blue and yellow for the Multiplicative Group and Additive Group, respectively. Error bars indicate standard error. * p<0.05, ** p<0.01, *** p<0.001.

The online version of this article includes the following figure supplement(s) for figure 5:

**Figure supplement 1.** The distractor effects shown in *Figure 5* remained significant after including an extra $HV + LV$ term in GLM3.

**Figure supplement 2.** The distractor effect between groups were maginally significant following the introduction of the $(HV + LV)\,T$ term to GLM3 and changing the $DV - HV$ terms to $DV$ to prevent collinearity between the regressors.

**Figure supplement 3.** The positive distractor effect in the Multiplicative Group shown in *Figure 5* remained significant after replacing the utility function based on the normative Expected Value model with values obtained by using the composite model.

to the GLM, because this term was included in some analyses of a previous study that used the same dataset (*Chau et al., 2020*). In the second analysis, we added an $(HV + LV)\,T$ term to the GLM. We noticed that this change led to inflation of the collinearity between the regressors, so we also replaced the (DV−HV) term by the DV term to mitigate the collinearity (*Figure 5—figure supplement 2*). In the third analysis, reported in *Figure 5—figure supplement 3*, we replaced the utility terms of GLM2. Since the above analyses involved using HV, LV, and DV values defined by the normative Expected Value model, here, we re-defined the values using the composite model prior to applying GLM2. Overall, in the Multiplicative Group a significant positive distractor effect was found in *Figure 5— figure supplement 1*. In the Additive Group a significant negative distractor effect was found in *Figure 5—figure supplements 1 and 3*. Crucially, all three analyses demonstrated the consistent trend that the distractor effects were different and opposite between the Multiplicative Group and the Additive Group.

## Discussion

It has been widely agreed that humans make irrational decisions in the presence of distractor options. However, there has been little consensus on how exactly distractors influence decision making. Does a seemingly attractive distractor impair or facilitate decision making? Does the distractor influence decision making at the level of overall utility or individual choice attributes? Often, one assumption behind these questions is that there is only a single type of distractor effect. Here, we demonstrated that, instead of a unidirectional effect, the way a distractor influences decision making depends on individuals' styles of integrating choice attributes during decision making. More specifically, those who employed a multiplicative style of integrating attributes were prone to a positive distractor effect. Conversely, those who employed an additive style of integrating attributes were prone to a negative distractor effect. These findings show that the precise way in which distractors affect decision making depends on an interaction between the distractor value and people's style of decision making.

At the neuroanatomical level, the negative distractor effect is mediated by the PPC, where signal modulation described by divisive normalization has been previously identified (*Chau et al., 2014*;

*Louie et al., 2011*). The same region is also crucial for perceptual decision making processes (*Shadlen and Shohamy, 2016*). The additive heuristics for combining choice attributes are closer to perceptual evaluation because distances in this subjective value space correspond linearly to differences in physical attributes of the stimuli, whereas normative (multiplicative) value has a non-linear relationship with them (*Figure 1c*). The same would apply even with a different choice of cues, as long as the information is conveyed by two independent visual features. It is well understood that many sensory mechanisms, such as those in primates' visual systems or fruit flies' olfactory systems, are subject to divisive normalization (*Carandini and Heeger, 2011*). Hence, the additive heuristics that are more closely based on sensory mechanisms could also be subject to divisive normalization, leading to negative distractor effects in decision making.

In contrast, the positive distractor effect is mediated by the mPFC (*Chau et al., 2014*; *Fouragnan et al., 2019*). Interestingly, the same or adjacent interconnected mPFC regions have also been linked to the mechanisms by which representational elements are integrated into new representations (*Barron et al., 2013*; *Klein-Flügge et al., 2022*; *Law et al., 2023*; *Papageorgiou et al., 2017*; *Schwartenbeck et al., 2023*). In a number of situations, such as multi-attribute decision making, understanding social relations, and abstract knowledge, the mPFC achieves this by using a spatial map representation characterised by a grid-like response (*Constantinescu et al., 2016*; *Bongioanni et al., 2021*; *Park et al., 2021*) and disrupting mPFC leads to the evaluation of composite choice options as linear functions of their components (*Bongioanni et al., 2021*). These observations suggest a potential link between positive distractor effects and mechanisms for evaluating multiple component options and this is consistent with the across-participant correlation that we observed between the strength of the positive distractor effect and the strength of non-additive (i.e. multiplicative) evaluation of the composite stimuli we used in the current task. Hence, one direction for model development may involve incorporating the idea that people vary in their ways of combining choice attributes and each way is susceptible to different types of distractor effect.

The current finding involves the use of a composite model that arbitrates between the additive and multiplicative strategies. While the additive strategy is a natural and simple approach for integrating non-interacting pieces of information, to some extent, participants also used the multiplicative strategy that was optimal in the current experiment. A general question for such composite models is whether people mix two strategies in a consistent manner on every trial or whether there is some form of probabilistic selection occurring between the two strategies on each trial such that only one strategy is used on any given trial while, on average, one strategy is more probable than the other. It would also be interesting to examine whether a composite model is appropriate in contexts where the optimal solution is additive or subtractive, such as those concerning quality and price. Further examination of these situations would require an independent estimation of subjective values in quantitative terms, such as by pupillometry or functional neuroimaging. Better understanding of this problem will also provide important insight into the precise way in which distractor effects operate at the single-trial level.

Other studies have provided evidence suggesting that multiple forms of distractor effects can co-exist. For example, when people were asked to choose between three food items, most people were poorer at choosing the best option when the worst option (i.e. the distractor) was still an appealing option (*Louie et al., 2013*; *Webb et al., 2020*). In other words, most people showed negative distractor effects, which is predicted by divisive normalization models (*Carandini and Heeger, 2011*; *Louie et al., 2013*). However, it is noticeable that the degree of negative distractor effect varies across individuals and some even showed the reverse, positive distractor effect. Similarly, it has been shown in social decision making that choice utility was best described by a divisive normalization model (*Chang et al., 2019*). However, at the same time, a positive distractor effect was found in choice behaviour, such that greater distractor values were instead associated with more choices of the best option, suggesting a positive distractor effect. The effect became even more robust when individual variability was considered in a mixed-effects model. Together, these results suggest that divisive normalization (which predicts negative distractor effects) and positive distractor effect may co-exist, but that they predominate in different aspects of decision making.

In the current study and in previous work, we have used or made reference to models of decision making that assume a noisy process of choice comparison occurs such as recurrent neural networks and drift diffusion models (*Shadlen and Shohamy, 2016*; *Wang, 2008*). Under this approach, positive

distractor effects are predicted when the comparison process becomes more accurate because of an impact on the noisy process of choice comparison (*Chau et al., 2020*; *Kohl et al., 2023*). However, it is worth noting that another class of models might assume that a choice representation itself is inherently noisy. According to this approach, on any given decision, a sample is drawn from a distribution of value estimates in a noisy representation of the option. Thus, when the representation is accessed, it might have a certain value on average but this value might vary from occasion to occasion. As a consequence, the value of a distractor that is 'drawn' during decision between two other options may be larger than the distractor's average value and may even exceed the value drawn from the less valuable choice option's distribution on the current trial. On such a trial, it may become especially clear that the better of the two options has a higher value than the alternative choice option. *Louie et al., 2013* suggest an explanation approximately along these lines when they reported a positive distractor effect during some decisions. Such different approaches share theoretical origins (*Shadlen and Shohamy, 2016*) and make related predictions about the impact of distractors on decision making.

Indeed, it is possible that multiple forms of distractor effects can co-exist because of their different neuroanatomical origins. In humans and monkeys, a positive distractor effect was found in decision signals in the mPFC (*Chau et al., 2014*; *Fouragnan et al., 2019*), whereas divisive normalization was found in decision signals in the PPC (*Chau et al., 2014*; *Louie et al., 2011*). As such, it should be expected that while these opposite distractor effects might sometimes diminish one another, disruption in one of the brain regions might result in the expression of the distractor effect related to the other brain region. Indeed, this idea is supported by empirical data showing that the parietal-related, negative distractor effect was more prominent in humans and monkeys with a lesion in the mPFC (*Noonan et al., 2010*; *Noonan et al., 2017*). In addition, the prefrontal-related, positive distractor effect was more prominent after the parietal cortex was transiently disrupted using transcranial magnetic stimulation (*Kohl et al., 2023*). These findings concur with the general notion that decision making is mediated by a distributed neural circuit, rather than a single, localised brain region.

## Methods

**Key resources table**

| Reagent type (species) or resource | Designation | Source or reference | Identifiers | Additional information |
|---|---|---|---|---|
| Software, algorithm | MATLAB R2022a | MATLAB | RRID:SCR_001622 | |

### Multi-attribute decision-making task and datasets

The current study re-analysed five published datasets of empirical data based on a multi-attribute decision-making task (*Chau et al., 2014*; *Gluth et al., 2018*). The experimental task involved participants making a decision between two options in the absence (Two-Option Trials) or presence (Distractor Trials) of a third distractor option that could not be chosen. During each trial, two or three stimuli associated with different reward magnitudes and probabilities (represented by colours and orientations) were randomly presented in selected screen quadrants (*Figure 1*). Immediately following stimulus onset (0.1 s), options that were available for selection were surrounded by orange boxes, while the distractor option that could not be selected was surrounded by a purple box. The choice of the participant was indicated by the change in colour of the surrounding box from orange to red. At the end of each trial, the edge of each stimulus changed to yellow if the choice was rewarded, and to grey if the choice was not rewarded. A total of 144 human participants were included in the analysis with data from the original fMRI dataset [N=21; (*Chau et al., 2014*)] and additional replication experiments (Experiments 1, 2, 3, and 4) performed by Gluth and colleagues [N=123; (*Gluth et al., 2018*)].

### GLM analysis of relative choice accuracy

The choice accuracy data from the Two-Option Trials and Distractor Trials were analysed separately using GLM1:

$$Logit\left(accuracy\right) = \beta_0 + \beta_1 z_{(HV-LV)} + \beta_2 z_{(DV-HV)} + \beta_3 z_{(HV-LV)} z_{(DV-HV)} + \varepsilon$$

where $HV$, $LV$, and $DV$ refer to the values of the chooseable higher value option, chooseable lower value option, and distractor, respectively. Here, values (except those in *Figure 5—figure supplement 3*) are defined as Expected Value (EV), calculated by multiplying magnitude and probability of reward.

$z_{(x)}$ refer to the z-scoring of term $x$, which were applied to all terms within the GLMs. The unexplained error is represented by $\varepsilon$. Trials in which the distractor or an empty quadrant were mistakenly chosen were excluded from the analysis. Fitting was performed using MATLAB's glmfit function by assuming a binomial linking function.

A binary variable ($T$) encoding trial type (i.e. 0=Two-Option Trials and 1=Distractor Trials) was introduced to combine both two-option and distractor trial types into a single GLM used to assess distractor-related interactions:

The choice accuracy data of the Two-Option Trials and Distractor Trials were analysed together using GLM2, which followed exactly the procedures described by *Cao and Tsetsos, 2022*:

$$Logit\left(accuracy\right) = \beta_0 + \beta_1 z_{(HV-LV)} + \beta_2 z_{(DV-HV)} + \beta_3 z_{(HV-LV)} z_{(DV-HV)} + \beta_4 T + \beta_5 z_{(HV-LV)} T +$$
$$\beta_6 z_{(DV-HV)} T + \beta_7 z_{(HV-LV)} z_{(DV-HV)} T + \varepsilon$$

We focused on the relative accuracy of Distractor Trials compared to Two-Option trials by considering the proportion of H choices among trials with H or L choices and excluding a small number of D-choice trials. We also identified matched Two-Option Trials for each distractor trial. The 150 Distractor Trials yielded 149 unique conditions exhibiting a unique combination of probability (P) and magnitude (X) attributes across the three options (H, L, and D). However, the 150 Two-Option Trials contained only 95 unique conditions carrying a unique combination of P and X across the two available options (H and L). As the distractor and Two-Option Trials do not exhibit 'one-to-one' mapping, some Distractor Trials will have more than one matched Two-Option Trials. The different counts of matched Two-Option Trials were used as 'observation weights' in the GLM. Through identifying all matched Two-Option Trials of every distractor trial, the baselining approach introduces 'trial-by-trial baseline accuracy' as a new dependent variable as described by *Cao and Tsetsos, 2022*.

## Computational modelling

To determine which model best describes participants' choice behaviour, we followed the procedures carried out by *Cao and Tsetsos, 2022*. We fitted three models to choice data from the control Two-Option Trials, namely the Expected Value (EV) model, Additive Utility (AU) model, and the Expected Value and Divisive Normalisation (EV +DN) model. In this study, we also included an additional composite (additive and multiplicative) model. For each model, we applied the softmax function as the basis for estimating choice probability:

$$p\left(choice = i\right) = \frac{e^{\vartheta U_i}}{\Sigma_{j=1}^2 e^{\vartheta U_j}}$$

where $U_i$ refers to the utility of option $i$ (either the HV or LV option) and $\vartheta$ refers to the inverse temperature parameter. Four models were used to estimate the options' utility $U_i$, based on their corresponding reward magnitude $X_i$ and probability $P_i$ (both rescaled to the interval between 0 and 1). The four models were the Expected Value model, Additive Utility model, Expected Value and Divisive normalisation model, and Composite model:

### Expected value (EV) model
This model employs a multiplicative rule for estimating utility:

$$U_i = X_i \times P_i$$

### Additive utility (AU) model
This model employs an additive rule for estimating utility based on the magnitude $X_i$ and probability $P_i$ as follows:

$$U_i = \gamma X_i + \left(1 - \gamma\right) P_i$$

where $\gamma$ is the magnitude/probability weighing ratio ($0 \leq \gamma \leq 1$).

## Expected value and divisive normalisation (EV+DN) model

We included the EV +DN model following the procedures carried out by *Cao and Tsetsos, 2022*. Compared to the EV models, here utilities were normalised by the inputs of all values as follows:

$$EV_i = X_i \times P_i$$

$$U_i = \frac{EV_i}{\Sigma_{j=1}^{N} EV_j}$$

where $EV_i$, $X_i$, and $P_i$ denote expected value, reward magnitude, and reward probability of option $i$, respectively.

## Composite model

We further explored the possibility that behaviour may be described as a mixture of both additive AU and multiplicative EV models (*Bongioanni et al., 2021*; *Scholl et al., 2014*):

$$U_i = \eta \left( X_i \times P_i \right) + \left( 1 - \eta \right) \left( \gamma X_i + \left( 1 - \gamma \right) P_i \right)$$

where $\gamma$ is the magnitude/probability weighing ratio ($0 \leq \gamma \leq 1$) and $\eta$ is the integration coefficient ($0 \leq \eta \leq 1$) determining the relative weighting of the additive and multiplicative value. Simulations reported by Bongioanni and colleagues (*Bongioanni et al., 2021*) prove that the composite model can be accurately discriminated from the simple additive and multiplicative models, even if the latter ones include non-linear distortions of the magnitude and probability dimensions. Additional simulations have shown that the fitted parameters can be recovered with high accuracy (i.e. with a high correlation between generative and recovered parameters).

## Model fitting and comparison

The empirical choice probabilities generated from the model predictions were used to calculate the binomial log-likelihood, following the procedures described by *Cao and Tsetsos, 2022*:

$$LL \propto p_e \log \left( p_m \right) + \left( 1 - p_e \right) \log \left( 1 - p_m \right)$$

here, $p_e$ and $p_m$ represent the empirical and model-predicted $p \left( H \, over \, L \right)$, respectively.

Model fitting was performed on the behavioural data to maximise the log-likelihood summed over trials. A grid of randomly generated starting values for free parameters of each model was used to fit each participant's data at least 10 times to avoid local optima. Model fitting was performed using MATLAB's fmincon function with the maximum number of function evaluations and iterations set to 5000 and the optimality and step tolerance set to $10^{-10}$. The variational Bayesian analysis (VBA) toolbox (*Daunizeau et al., 2014*; *Rigoux et al., 2014*) was used to calculate each model's posterior frequency and protected exceedance probability. Aligning with the methods used by *Cao and Tsetsos, 2022*, only the trials where H or L responses were made were included in the analysis. Trials in which participants opted for the unchooseable D were excluded.

Fivefold cross-validation was performed for model comparison. First, this involved dividing the 150 trials for each participant into five folds. Four random folds of trials would be randomly selected to generate best-fitting parameters used to calculate the log-likelihood summed across trials in the unchosen fold. We repeated this process five times for each unchosen fold and computed the average log-likelihood across cross-validation folds to generate the cross-validated log-likelihood for each model. We then simulated each model's behaviour using the best-fitting parameters obtained during model fitting. The simulated behaviour was cross-fitted to all models to calculate the log-likelihoods summed over trials. Lastly, the goodness-of-fit for the models were evaluated using Bayesian model comparison.

## Acknowledgements

This work was supported by the Hong Kong Research Grants Council (15105522), the Wellcome Trust grant (221794/Z/20/Z), and the Projects of Strategic Importance of The Hong Kong Polytechnic University (1-ZE0E).

# Additional information

## Funding

| Funder | Grant reference number | Author |
|---|---|---|
| Research Grants Council, University Grants Committee | 15105522 | Bolton KH Chau |
| Wellcome Trust | 10.35802/221794 | Matthew FS Rushworth |
| The Hong Kong Polytechnic University | Projects of Strategic Importance 1-ZE0E | Bolton KH Chau |

The funders had no role in study design, data collection and interpretation, or the decision to submit the work for publication. For the purpose of Open Access, the authors have applied a CC BY public copyright license to any Author Accepted Manuscript version arising from this submission.

## Author contributions

Jing Jun Wong, Formal analysis, Methodology, Writing – original draft, Writing – review and editing; Alessandro Bongioanni, Methodology, Writing – review and editing; Matthew FS Rushworth, Funding acquisition, Methodology, Writing – review and editing; Bolton KH Chau, Conceptualization, Resources, Supervision, Funding acquisition, Methodology, Writing – original draft, Project administration, Writing – review and editing

## Author ORCIDs

Jing Jun Wong ⓘ https://orcid.org/0000-0001-5241-6997
Alessandro Bongioanni ⓘ http://orcid.org/0000-0002-4996-6098
Matthew FS Rushworth ⓘ http://orcid.org/0000-0002-5578-9884
Bolton KH Chau ⓘ https://orcid.org/0000-0002-6854-5176

## Ethics

This study was approved by the ethics committee of The Hong Kong Polytechnic University.

Reviewer #1 (Public review): https://doi.org/10.7554/eLife.91102.3.sa1
Reviewer #2 (Public review): https://doi.org/10.7554/eLife.91102.3.sa2
Reviewer #3 (Public review): https://doi.org/10.7554/eLife.91102.3.sa3
Author response https://doi.org/10.7554/eLife.91102.3.sa4

# Additional files

## Supplementary files

• MDAR checklist

## Data availability

The current manuscript re-analysed five published datasets of empirical data, so no new data have been generated for this manuscript. The code essential for replicating the main findings of the study can be downloaded at https://github.com/JingJun-Wong/distractor-effect (copy archived by *Wong, 2024*).

The following previously published datasets were used:

| Author(s) | Year | Dataset title | Dataset URL | Database and Identifier |
|---|---|---|---|---|
| Chau BKH, Kolling N, Hunt LT, Walton ME, Rushworth MFS | 2014 | A neural mechanism underlying failure of optimal choice with multiple alternatives | https://doi.org/10.5061/dryad.040h9t7 | Dryad Digital Repository, 10.5061/dryad.040h9t7 |

*Continued on next page*

*Continued*

| Author(s) | Year | Dataset title | Dataset URL | Database and Identifier |
|---|---|---|---|---|
| Gluth S, Spektor MS, Rieskamp J | 2018 | Value-based attentional capture affects multi-alternative decision making | https://osf.io/8r4fh/ | Open Science Framework, 8r4fh |

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
