## [Editor Report · eLife assessment]

This manuscript provides a **valuable** demonstration that distractor effects in multi-attribute decision-making correlate with the form of attribute integration (additive vs. multiplicative). The evidence supporting the conclusions is **convincing**, but there are questions about how to interpret the findings. The manuscript will be interesting to decision-making researchers in neuroscience, psychology, and related fields.

---

## [Referee Report · Reviewer #1 (Public review)]

Summary:

The current study provided a follow-up analysis using published datasets focused on the individual variability of both the distraction effect (size and direction) and the attribute integration style, as well as the association between the two. The authors tried to answer the question of whether the multiplicative attribute integration style concurs with a more pronounced and positively oriented distraction effect.

Strengths:

The analysis extensively examined the impacts of various factors on decision accuracy, with particular focus on using two-option trials as control trials, following the approach established by Cao & Tsetsos (2022). The statistical significance results were clearly reported.

The authors meticulously conducted supplementary examinations, incorporating the additional term HV+LV into GLM3. Furthermore, they replaced the utility function from the expected value model with values from the composite model.

Weaknesses:

The authors did a great job addressing the weaknesses I raised in the previous round of review, except on the generalizability of the current result in the larger context of multi-attribute decision-making. It is not really a weakness of the manuscript but more of a limitation of the studied topic, so I want to keep this comment for public readers.

The reward magnitude and probability information are displayed using rectangular bars of different colors and orientations. Would that bias subjects to choose an additive rule instead of the multiplicative rule? Also, could the conclusion be extended to other decision contexts such as quality and price, where a multiplicative rule is hard to formulate?

Overall, the authors have achieved their aims after clarifying that the study was trying to establish a correlation between the integration style and attraction effect. This result may be useful to inspire neuroimaging or neuromodulation studies that investigate multi-attribute decision making.

---

## [Referee Report · Reviewer #2 (Public review)]

This paper addresses the empirical demonstration of "distractor effects" in multi-attribute decision-making. It continues a debate in the literature on the presence (or not) of these effects, which domains they arise in, and their heterogeneity across subjects. The domain of the study is in a particular type of multi-attribute decision-making: choices over risky lotteries. The paper reports a re-analysis of lottery data from multiple experiments run previously by the authors and other labs involved in the debate.

Methodologically, the analysis assumes a number of simple forms for how attributes are aggregated (adaptively, or multiplicatively, or both) and then applies a "reduced form" logistic regression to the choices with a number of interaction terms intended to control for various features of the choice set. One of these interactions, modulated by ternary/binary treatment, is interpreted as a "distractor effect."

The claimed contribution of the re-analysis is to demonstrate correlation in the strength/sign of this treatment effect with another estimated parameter: the relative mixture of additive/multiplicative preferences.

Major Issues

(1) How to Interpret GLM 1 and 2

This paper, and others before it, have used a binary logistic regression with a number of interaction terms to attempt to control for various features of the choice set and how they influence choice. It is important to recognize that this modelling approach is not derived from a theoretical claim about the form of the computational model that guides decision-making in this task, nor an explicit test for a distractor effect. This can be seen most clearly in the equations after line 321 and its corresponding log-likelihood after 354, which contain no parameter or test for "distractor effects". Rather the computational model assumes a binary choice probability, and then shoehorns the test for distractor effects via a binary/ternary treatment interaction in a separate regression (GLM 1 and 2). This approach has already led to multiple misinterpretations in the literature (see Cao & Tsetsos, 2022; Webb et al., 2020). One of these misinterpretations occurred in the datasets the authors study, in which the lottery stimuli contained a confound with the interaction that Chau et al., (2014) were interpreting as a distractor effect (GLM 1). Cao & Tsetsos (2022) demonstrated that the interaction was significant in binary choice data from the study, therefore it can not be caused by a third alternative. This paper attempts to address this issue with a further interaction with the binary/ternary treatment (GLM 2). Therefore the difference in the interaction across the two conditions is claimed to now be the distractor effect. The validity of this claim brings us to what exactly is meant by a "distractor effect."

The paper begins by noting that "Rationally, choices ought to be unaffected by distractors" (line 33). This is not true. There are many normative models which allow for the value of alternatives (even low-valued "distractors") to influence choices, including a simple random utility model. Since Luce (1959), it has been known that the axiom of "Independence of Irrelevant Alternatives" (that the probability ratio between any two alternatives not depend on a third) is an extremely strong axiom, and only a sufficiency axiom for a random utility representation (Block and Marschak, 1959). It is not a necessary condition of a utility representation, and if this is our definition of rational (which is highly debatable), not necessary for it either. Countless empirical studies have demonstrated that IIA is falsified, and a large number of models can address it, including a simple random utility model with independent normal errors (i.e. a multivariate Probit model). In fact, it is only the multinomial Logit model that imposes IIA. It is also why so much attention is paid to the asymmetric dominance effect, which is a violation of a necessary condition for random utility (the Regularity axiom).

So what do the authors even mean by a "distractor effect." It is true that the form of IIA violations (i.e. their path through the probability simplex as the low-option varies) tells us something about the computational model underlying choice (after all, different models will predict different patterns). But we do not know how the interaction terms in the binary logit regression relate to the pattern of the violations because there is no formal theory that relates them. Any test for relative value coding is a joint test of the computational model and the form of the stochastic component (Webb et al,. 2020). These interaction terms may simply be picking up substitution patterns that can be easily reconciled with some form of random utility. While we can not check all forms of random utility in these datasets (because the class of such models is large), this paper doesn't even rule any of these models out.

(2) How to Interpret the Composite (Mixture) model?

On the other side of the correlation is the results from the mixture model for how decision-makers aggregate attributes. The authors report that most subjects are best represented by a mixture between additive and multiplicative aggregation models. The authors justify this with the proposal that these values are computed in different brain regions and then aggregated (which is reasonable, though raises the question of "where" if not the mPFC). But an equally reasonable interpretation is that the improved fit of the mixture model simply reflects a misspecification of two extreme aggregation process (additive and EV), so the log-likelihood is maximized at some point in between them.

One possibility is a model with utility curvature. How much of this result is just due to curvature in valuation? There are many reasonable theories for why we should expect curvature in utility for human subjects (for example, limited perception: Robson, 2001, Khaw, Li Woodford, 2019; Netzer et al., 2022) and of course many empirical demonstrations of risk aversion for small stakes lotteries. The mixture model, on the other hand, has parametric flexibility.

There is also a large literature on testing expected utility jointly with stochastic choice, and the impact of these assumptions on parameter interpretation (Loomes & Sugden, 1998; Apesteguia & Ballester, 2018; Webb, 2019). This relates back to the point above: the mixture may reflect the joint assumption of how choice departs from deterministic EV.

(3) So then how should we interpret the correlation that the authors report?

On one side we have the impact of the binary/ternary treatment which demonstrates some impact of the low value alternative on a binary choice probability. This may reflect some deep flaw in existing theories of choice, or it may simply reflect some departure from purely deterministic expected value maximization that existing theories can address. We have no theory to connect it to, so we cannot tell. On the other side of the correlation with have the mixture between additive and multiplicative preferences over risk. This result may reflect two distinct neural processes at work, or it may simply reflect a misspecification of the manner in which humans perceive and aggregate attributes of a lottery (or even just the stimuli in this experiment) by these two extreme candidates (additive vs. EV). Again, this would entail some departure from purely deterministic expected value maximization that existing theories can address.

It is entirely possible that the authors are reporting a result that points to the more exciting of these two possibilities. But it is also possible (and perhaps more likely) that the correlation is more mundane. The paper does not guide us to theories that predict such a correlation, nor reject any existing ones. In my opinion, we should be striving for theoretically-driven analyses of datasets, where the interpretation of results is clearer.

(4) Finally, the results from these experiments might not have external validity for two reasons. First, the normative criterion for multi-attribute decision-making differs depending on whether the attributes are lotteries or nor (i.e. multiplicative vs additive). Whether it does so for humans is a matter of debate. Therefore if the result is unique to lotteries, it might not be robust for multi-attribute choice more generally. The paper largely glosses over this difference and mixes literature from both domains. Second, the lottery information was presented visually and there is literature suggesting this form of presentation might differ from numerical attributes. Which is more ecologically valid is also a matter of debate.

Minor Issues:

The definition of EV as a normative choice baseline is problematic. The analysis requires that EV is the normative choice model (this is why the HV-LV gap is analyzed and the distractor effect defined in relation to it). But if the binary/ternary interaction effect can be accounted for by curvature of a value function, this should also change the definition of which lottery is HV or LV for that subject!

Comments on latest version: the authors did respond to some of my comments with discussion points in the paper.

References

Apesteguia, J. & Ballester, M. Monotone stochastic choice models: The case of risk and time preferences. Journal of Political Economy (2018).

Block, H. D. & Marschak, J. Random Orderings and Stochastic Theories of Responses. Cowles Foundation Discussion Papers (1959).

Khaw, M. W., Li, Z. & Woodford, M. Cognitive Imprecision and Small-Stakes Risk Aversion. Rev. Econ. Stud. 88, 1979-2013 (2020).

Loomes, G. & Sugden, R. Testing Different Stochastic Specifications of Risky Choice. Economica 65, 581-598 (1998).

Luce, R. D. Indvidual Choice Behaviour. (John Wiley and Sons, Inc, 1959).

Netzer, N., Robson, A. J., Steiner, J. & Kocourek, P. Endogenous Risk Attitudes. SSRN Electron. J. (2022) doi:10.2139/ssrn.4024773.

Robson, A. J. Why would nature give individuals utility functions? Journal of Political Economy 109, 900-914 (2001).

Webb, R. The (Neural) Dynamics of Stochastic Choice. Manage Sci 65, 230-255 (2019).

---

## [Referee Report · Reviewer #3 (Public review)]

Summary:

The way an unavailable (distractor) alternative impacts decision quality is of great theoretical importance. Previous work, led by some of the authors of this study, had converged on a nuanced conclusion wherein the distractor can both improve (positive distractor effect) and reduce (negative distractor effect) decision quality, contingent upon the difficulty of the decision problem. In very recent work, Cao and Tsetsos (2022) reanalyzed all relevant previous datasets and showed that once distractor trials are referenced to binary trials (in which the distractor alternative is not shown to participants), distractor effects are absent. Cao and Tsetsos further showed that human participants heavily relied on additive (and not multiplicative) integration of rewards and probabilities.

The present study by Wong et al. puts forward a novel thesis according to which interindividual differences in the way of combining reward attributes underlie the absence of detectable distractor effect at the group level. They re-analysed the 144 human participants and classified participants into a "multiplicative integration" group and an "additive integration" group based on a model parameter, the "integration coefficient", that interpolates between the multiplicative utility and the additive utility in a mixture model. They report that participants in the "multiplicative" group show a negative distractor effect while participants in the "additive" group show a positive distractor effect. These findings are extensively discussed in relation to the potential underlying neural mechanisms.

Strengths:

- The study is forward looking, integrating previous findings well, and offering a novel proposal on how different integration strategies can lead to different choice biases.

- The authors did an excellent job in connecting their thesis with previous neural findings. This is a very encompassing perspective that is likely to motivate new studies towards better understanding of how humans and other animals integrate information in decisions under risk and uncertainty.

- Despite that some aspects of the paper are very technical, methodological details are well explained and the paper is very well written.

Weaknesses:

- The authors quantify the distractor variable as "DV - HV", i.e., the relative distractor variable. Conclusions mostly hold when the distractor is quantified in absolute terms (as "DV", see also Cao & Tsetsos, 2023). However, it is not entirely clear why the impact of the distractor alternative is not identical when the distractor variable is quantified in absolute vs. relative terms. Although understanding this nuanced point seems to extend beyond the scope of the paper, it could provide valuable decision-theoretic (and mechanistic) insights.

- The central finding of this study is that participants who integrate reward attributes multiplicatively show a positive distractor effect while participants who integrate additively show a negative distractor effect. This is a very interesting and intriguing observation. However, it does not explain why the integration strategy covaries with the direction of the distractor effect. As the authors acknowledge, the composite model is not explanatory. Although beyond the scope of this paper, it would be valuable to provide a mechanistic explanation of this covariation pattern.

---

## [Author Response]

The following is the authors’ response to the current reviews.

**Reviewer #1 (Public Review):**
The authors did a great job addressing the weaknesses I raised in the previous round of review, except on the generalizability of the current result in the larger context of multi-attribute decision-making. It is not really a weakness of the manuscript but more of a limitation of the studied topic, so I want to keep this comment for public readers.The reward magnitude and probability information are displayed using rectangular bars of different colors and orientations. Would that bias subjects to choose an additive rule instead of the multiplicative rule? Also, could the conclusion be extended to other decision contexts such as quality and price, where a multiplicative rule is hard to formulate?

We thank the reviewer for the comment. With regards whether the current type of stimuli may have biased participants to use an additive rule rather, we believe many other forms of stimuli for representing choice attributes would be equally likely to cause a similar bias. This is because the additive strategy is an inherently simplistic and natural way to integrate different pieces of non-interacting information. More importantly, even though it is easy to employ an additive strategy, most participants still demonstrated some levels of employing the multiplicative rule. However, it would indeed be interesting for future studies to explore whether the current composite model remains dominant in situations where the optimal solutions require an additive or subtractive rule, such as those concerning quality and price.

“The same would apply even with a different choice of cues as long as the information is conveyed by two independent visual features.”

“While the additive strategy is a natural and simple approach for integrating non-interacting pieces of information, to some extent, participants also used the multiplicative strategy that was optimal in the current experiment. A general question for such composite models is whether people mix two strategies in a consistent manner on every trial or whether there is some form of probabilistic selection occurring between the two strategies on each trial such that only one strategy is used on any given trial while, on average, one strategy is more probable than the other. It would also be interesting to examine whether a composite model is appropriate in contexts where the optimal solution is additive or subtractive, such as those concerning quality and price.”

The following is the authors’ response to the original reviews.

**Reviewer #1 (Public Review):**
Summary:The current study provided a follow-up analysis using published datasets focused on the individual variability of both the distraction effect (size and direction) and the attribute integration style, as well as the association between the two. The authors tried to answer the question of whether the multiplicative attribute integration style concurs with a more pronounced and positively oriented distraction effect.Strengths:The analysis extensively examined the impacts of various factors on decision accuracy, with a particular focus on using two-option trials as control trials, following the approach established by Cao & Tsetsos (2022). The statistical significance results were clearly reported.The authors meticulously conducted supplementary examinations, incorporating the additional term HV+LV into GLM3. Furthermore, they replaced the utility function from the expected value model with values from the composite model.

We thank the reviewer for the positive response and are pleased that the reviewer found our report interesting.

Reviewer #1 Comment 1Weaknesses:There are several weaknesses in terms of theoretical arguments and statistical analyses.First, the manuscript suggests in the abstract and at the beginning of the introduction that the study reconciled the "different claims" about "whether distraction effect operates at the level of options' component attributes rather than at the level of their overall value" (see line 13-14), but the analysis conducted was not for that purpose. Integrating choice attributes in either an additive or multiplicative way only reflects individual differences in combining attributes into the overall value. The authors seemed to assume that the multiplicative way generated the overall value ("Individuals who tended to use a multiplicative approach, and hence focused on overall value", line 20-21), but such implicit assumption is at odds with the statement in line 77-79 that people may use a simpler additive rule to combine attributes, which means overall value can come from the additive rule.

We thank the reviewer for the comment. We have made adjustments to the manuscript to ensure that the message delivered within this manuscript is consistent. Within this manuscript, our primary focus is on the different methods of value integration in which the overall value is computed (i.e., additive, multiplicative, or both), rather than the interaction at the individual level of attributes. However, we do not exclude the possibility that the distractor effect may occur at multiple levels. Nevertheless, in light of the reviewer’s comment, we agree that we should focus the argument on whether distractors facilitate or impair decision making and downplay the separate argument about the level at which distractor effects operate. We have now revised the abstract:

“It is widely agreed that people make irrational decisions in the presence of irrelevant distractor options. However, there is little consensus on whether decision making is facilitated or impaired by the presence of a highly rewarding distractor or whether the distraction effect operates at the level of options’ component attributes rather than at the level of their overall value. To reconcile different claims, we argue that it is important to incorporate consideration of the diversity of people’s ways of decision making. We focus on a recent debate over whether people combine choice attributes in an additive or multiplicative way. Employing a multi-laboratory dataset investigating the same decision making paradigm, we demonstrated that people used a mix of both approaches and the extent to which approach was used varied across individuals. Critically, we identified that this variability was correlated with the effect of the distractor on decision making. Individuals who tended to use a multiplicative approach to compute value, showed a positive distractor effect. In contrast, in individuals who tended to use an additive approach, a negative distractor effect (divisive normalisation) was prominent. These findings suggest that the distractor effect is related to how value is constructed, which in turn may be influenced by task and subject specificities. Our work concurs with recent behavioural and neuroscience findings that multiple distractor effects co-exist.” (Lines 12-26)

Furthermore, we acknowledge that the current description of the additive rule could be interpreted in several ways. The current additive utility model described as:Ui=γXi+(1−γ)Pi

where is the options’ utility, is the reward magnitude, is the probability, and is the magnitude/probability weighing ratio . If we perform comparison between values according to this model (i.e., HV against LV), we would arrive at the following comparison:UHV=γXHV+(1−γ)PHVULV=γXLV+(1−γ)PLV

UHV−ULV=[γXHV+(1−γ)PHV]−[γXLV+(1−γ)PLV]

If we rearrange (1), we will arrive at:UHV−ULV=[γXHV−γXLV]+[(1−γ)PHV−(1−γ)PLV]

While equations (1) and (2) are mathematically equivalent, equation (1) illustrates the interpretation where the comparison of the utilities occurs after value integration and forming an overall value. On the other hand, equation (2) can be broadly interpreted as the comparison of individual attributes in the absence of an overall value estimate for each option. Nonetheless, while we do not exclude the possibility that the distractor effect may occur at multiple levels, we have made modifications to the main manuscript employ more consistently a terminology referring to different methods of value estimation while recognizing that our empirical results are compatible with both interpretations.

Reviewer #1 Comment 2The second weakness is sort of related but is more about the lack of coherent conceptual understanding of the "additive rule", or "distractor effect operates at the attribute level". In an assertive tone (lines 77-80), the manuscript suggests that a weighted sum integration procedure of implementing an "additive rule" is equal to assuming that people compare pairs of attributes separately, without integration. But they are mechanistically distinct. The additive rule (implemented using the weighted sum rule to combine probability and magnitude within each option and then applying the softmax function) assumes value exists before comparing options. In contrast, if people compare pairs of attributes separately, preference forms based on the within-attribute comparisons. Mathematically these two might be equivalent only if no extra mechanisms (such as inhibition, fluctuating attention, evidence accumulation, etc) are included in the within-attribute comparison process, which is hardly true in the three-option decision.

We thank the reviewer for the comment. As described in our response to Reviewer #1 Comment 1, we are aware and acknowledge that there may be multiple possible interpretations of the additive rule. We also agree with the reviewer that there may be additional mechanisms that are involved in three- or even two- option decisions, but these would require additional studies to tease apart. Another motivation for the approach used here, which does not explicitly model the extra mechanisms the reviewer refers to was due to the intention of addressing and integrating findings from previous studies using the same dataset [i.e. (Cao & Tsetsos, 2022; Chau et al., 2020)]. Lastly, regardless of the mechanistic interpretation, our results show a systematic difference in the process of value estimation. Modifications to the manuscript text have been made consistent with our motivation (please refer to the reply and the textual changes proposed in response to the reviewer’s previous comment: Reviewer #1 Comment 1).

Reviewer #1 Comment 3Could the authors comment on the generalizability of the current result? The reward magnitude and probability information are displayed using rectangular bars of different colors and orientations. Would that bias subjects to choose an additive rule instead of the multiplicative rule? Also, could the conclusion be extended to other decision contexts such as quality and price, whether a multiplicative rule is hard to formulate?

We thank the reviewer for the comment. We agree with the observation that the stimulus space, with colour linearly correlated with magnitude, and orientation linearly correlated with probability, may bias subjects towards an additive rule. But that’s indeed the point: in order to maximise reward, subjects should have focused on the outcome space without being driven by the stimulus space. In practice, people are more or less successful in such endeavour. Nevertheless, we argue that the specific choice of visual stimuli we used is no more biased towards additive space than any other. In fact, as long as two or more pieces of information are provided for each option, as opposed to a single cue whose value was previously learned, there will always be a bias towards an additive heuristic (a linear combination), regardless of whether the cues are shapes, colours, graphs, numbers, words.

As the reviewer suggested, the dataset analyzed in the current manuscript suggests that the participants were leaning towards the additive rule. Although there was a general tendency using the additive rule while choosing between the rectangular bars, we can still observe a spread of individuals using either, or both, additive and multiplicative rules, suggesting that there was indeed diversity in participants’ decision making strategies in our data.

In previous studies, it was observed that human and non-human individuals used a mix of multiplicative and additive rules when they were tested on experimental paradigms different from ours (Bongioanni et al., 2021; Farashahi et al., 2019; Scholl et al., 2014). It was also observed that positive and negative distractor effects can be both present in the same data set when human and non-human individuals made decisions about food and social partner (Chang et al., 2019; Louie et al., 2013). It was less clear in the past whether the precise way a distractor affects decision making (i.e., positive/negative distractor effect) is related to the use of decision strategy (i.e., multiplicative/additive rules) and this is exactly what we are trying to address in this manuscript. A follow-up study looking at neural data (such as functional magnetic resonance imaging data) could provide a better understanding of the mechanistic nature of the relationship between distractor effects and decision strategy that we identified here.

We agree with the reviewer that it is true that a multiplicative strategy may not be applicable to some decision contexts. Here it is important to look at the structure of the optimal solution (the one maximizing value in the long run). Factors modulating value (such as probability and temporal delay) require a non-linear (e.g., multiplicative solution), while factors of the cost-benefit form (such as effort and price) require a linear solution (e.g., subtraction). In the latter scenario the additive heuristic would coincide with the optimal solution, and the effect addressed in this study may not be revealed. Nonetheless, the present data supports the notion of distinct neural mechanisms at least for probabilistic decision-making, and is likely applicable to decision-making in general.

Our findings, in conjunction with the literature, also suggest that a positive distractor effect could be a general phenomenon in decision mechanisms that involve the medial prefrontal cortex. For example, it has been shown that the positive distractor effect is related to a decision mechanism linked to medial prefrontal cortex [especially the ventromedial prefrontal cortex (Chau et al., 2014; Noonan et al., 2017)]. It is also known a similar brain region is involved not only when individuals are combining information using a multiplicative strategy (Bongioanni et al., 2021), but also when they are combining information to evaluate new experience or generalize information (Baram et al., 2021; Barron et al., 2013; Park et al., 2021). We have now revised the Discussion to explain this:

“In contrast, the positive distractor effect is mediated by the mPFC (Chau et al., 2014; Fouragnan et al., 2019). Interestingly, the same or adjacent, interconnected mPFC regions have also been linked to the mechanisms by which representational elements are integrated into new representations (Barron et al., 2013; Klein-Flügge et al., 2022; Law et al., 2023; Papageorgiou et al., 2017; Schwartenbeck et al., 2023). In a number of situations, such as multi-attribute decision making, understanding social relations, and abstract knowledge, the mPFC achieves this by using a spatial map representation characterised by a grid-like response (Constantinescu et al., 2016; Bongioanni et al., 2021; Park et al., 2021) and disrupting mPFC leads to the evaluation of composite choice options as linear functions of their components (Bongioanni et al., 2021). These observations suggest a potential link between positive distractor effects and mechanisms for evaluating multiple component options and this is consistent with the across-participant correlation that we observed between the strength of the positive distractor effect and the strength of non-additive (i.e., multiplicative) evaluation of the composite stimuli we used in the current task. Hence, one direction for model development may involve incorporating the ideas that people vary in their ways of combining choice attributes and each way is susceptible to different types of distractor effect.” (Lines 260-274)

Reviewer #1 Comment 4The authors did careful analyses on quantifying the "distractor effect". While I fully agree that it is important to use the matched two-option trials and examine the interaction terms (DV-HV)T as a control, the interpretation of the results becomes tricky when looking at the effects in each trial type. Figure 2c shows a positive DV-HV effect in two-option trials whereas the DV-HV effect was not significantly stronger in three-option trials. Further in Figure 5b,c, in the Multiplicative group, the effect of DV-HV was absent in the two-option trials and present in the three-option trials. In the Additive group, however, the effect of DV-HV was significantly positive in the two-option trials but was significantly lowered in the three-option trials. Hence, it seems the different distractor effects were driven by the different effects of DV-HV in the two-option trials, rather than the three-option trials?

We thank the reviewer for the comment. While it may be a bit more difficult to interpret, the current method of examining the (DV−HV)T term rather than (DV−HV) term was used because it was the approach used in a previous study (Cao & Tsetsos, 2022).

During the design of the original experiments, trials were generated pseudo-randomly until the DV was sufficiently decorrelated from HV−LV. While this method allows for better group-level examination of behaviour, Cao and Tsetsos were concerned that this approach may have introduced unintended confounding covariations to some trials. In theory, one of the unintended covariations could occur between the DV and specific sets of reward magnitude and probability of the HV and LV. The covariation between parameters can lead to an observable positive distractor effect in the DV−HV as a consequence of the attraction effect or an unintended byproduct of using an additive method of integrating attributes [for further elaboration, please refer to Figure 1 in (Cao & Tsetsos, 2022)]. While it may have some limitations, the approach suggested by Cao and Tsetsos has the advantage of leveraging the DV−HV term to absorb any variance contributed by possible confounding factors such that true distractor effects, if any, can be detected using the (DV−HV)T term.

Reviewer #1 Comment 5Note that the pattern described above was different in Supplementary Figure 2, where the effect of DV-HV on the two-option trials was negative for both Multiplicative and Additive groups. I would suggest considering using Supplementary Figure 2 as the main result instead of Figure 5, as it does not rely on multiplicative EV to measure the distraction effect, and it shows the same direction of DV-HV effect on two-option trials, providing a better basis to interpret the (DV-HV)T effect.

We thank the reviewer for the comments and suggestion. However, as mentioned in the response to Reviewer #1 Comment 4, the current method of analysis adopted in the manuscript and the interpretation of only (DV−HV)T is aimed to address the possibility that the (DV−HV) term may be capturing some confounding effects due to covariation. Given that the debate that is addressed specifically concerns the (DV−HV)T term, we elected to display Figure 5 within the main text and keep the results of the regression after replacing the utility function with the composite model as Supplementary Figure 5 (previously labelled as Supplementary Figure 2).

**Reviewer #2 (Public Review):**
This paper addresses the empirical demonstration of "distractor effects" in multi-attribute decision-making. It continues a debate in the literature on the presence (or not) of these effects, which domains they arise in, and their heterogeneity across subjects. The domain of the study is a particular type of multi-attribute decision-making: choices over risky lotteries. The paper reports a re-analysis of lottery data from multiple experiments run previously by the authors and other laboratories involved in the debate.Methodologically, the analysis assumes a number of simple forms for how attributes are aggregated (adaptively, multiplicatively, or both) and then applies a "reduced form" logistic regression to the choices with a number of interaction terms intended to control for various features of the choice set. One of these interactions, modulated by ternary/binary treatment, is interpreted as a "distractor effect."The claimed contribution of the re-analysis is to demonstrate a correlation in the strength/sign of this treatment effect with another estimated parameter: the relative mixture of additive/multiplicative preferences.

We thank the reviewer for the positive response and are pleased that the reviewer found our report interesting.

Reviewer #2 Comment 1Major Issues(1) How to Interpret GLM 1 and 2This paper, and others before it, have used a binary logistic regression with a number of interaction terms to attempt to control for various features of the choice set and how they influence choice. It is important to recognize that this modelling approach is not derived from a theoretical claim about the form of the computational model that guides decision-making in this task, nor an explicit test for a distractor effect. This can be seen most clearly in the equations after line 321 and its corresponding log-likelihood after 354, which contain no parameter or test for "distractor effects". Rather the computational model assumes a binary choice probability and then shoehorns the test for distractor effects via a binary/ternary treatment interaction in a separate regression (GLM 1 and 2). This approach has already led to multiple misinterpretations in the literature (see Cao & Tsetsos, 2022; Webb et al., 2020). One of these misinterpretations occurred in the datasets the authors studied, in which the lottery stimuli contained a confound with the interaction that Chau et al., (2014) were interpreting as a distractor effect (GLM 1). Cao & Tsetsos (2022) demonstrated that the interaction was significant in binary choice data from the study, therefore it can not be caused by a third alternative. This paper attempts to address this issue with a further interaction with the binary/ternary treatment (GLM 2). Therefore the difference in the interaction across the two conditions is claimed to now be the distractor effect. The validity of this claim brings us to what exactly is meant by a "distractor effect."The paper begins by noting that "Rationally, choices ought to be unaffected by distractors" (line 33). This is not true. There are many normative models that allow for the value of alternatives (even low-valued "distractors") to influence choices, including a simple random utility model. Since Luce (1959), it has been known that the axiom of "Independence of Irrelevant Alternatives" (that the probability ratio between any two alternatives does not depend on a third) is an extremely strong axiom, and only a sufficiency axiom for a random utility representation (Block and Marschak, 1959). It is not a necessary condition of a utility representation, and if this is our definition of rational (which is highly debatable), not necessary for it either. Countless empirical studies have demonstrated that IIA is falsified, and a large number of models can address it, including a simple random utility model with independent normal errors (i.e. a multivariate Probit model). In fact, it is only the multinomial Logit model that imposes IIA. It is also why so much attention is paid to the asymmetric dominance effect, which is a violation of a necessary condition for random utility (the Regularity axiom).So what do the authors even mean by a "distractor effect." It is true that the form of IIA violations (i.e. their path through the probability simplex as the low-option varies) tells us something about the computational model underlying choice (after all, different models will predict different patterns). However we do not know how the interaction terms in the binary logit regression relate to the pattern of the violations because there is no formal theory that relates them. Any test for relative value coding is a joint test of the computational model and the form of the stochastic component (Webb et al, 2020). These interaction terms may simply be picking up substitution patterns that can be easily reconciled with some form of random utility. While we can not check all forms of random utility in these datasets (because the class of such models is large), this paper doesn't even rule any of these models out.

We thank the reviewer for the comment. In this study, one objective is to address an issue raised by Cao and Tsetsos (2022), suggesting that the distractor effect claimed in the Chau et al. (2014) study was potentially confounded by unintended correlation introduced between the distractor and the chooseable options. They suggested that this could be tested by analyzing the control binary trials and the experimental ternary trials in a single model (i.e., GLM2) and introducing an interaction term (DV−HV)T. The interaction term can partial out any unintended confound and test the distractor effect that was present specifically in the experimental ternary trials. We adopted these procedures in our current studies and employed the interaction term to test the distractor effects. The results showed that overall there was no significant distractor effect in the group. We agree with the reviewer’s comment that if we were only analysing the ternary trials, a multinomial probit model would be suitable because it allows noise correlation between the choices. Alternatively, had a multinomial logistic model been applied, a Hausman-McFadden Test could be run to test whether the data violates the assumption of independence of irrelevant alternatives (IIA). However, in our case, a binomial model is preferred over a multinomial model because of: (1) the inclusion of the binary trials, and (2) the small number of trials in which the distractor was chosen (the median was 4% of all ternary trials).

However, another main objective of this study is to consider the possibility that the precise distractor effect may vary across individuals. This is exactly why we employed the composite model to estimate individual’s decision making strategy and investigated how that varied with the precise way the distractor influenced decision making.

In addition, we think that the reviewer here is raising a profound point and one with which we are in sympathy; it is true that random noise utility models can predict deviations from the IIA axiom. Central to these approaches is the notion that the representations of the values of choice options are noisy. Thus, when the representation is accessed, it might have a certain value on average but this value might vary from occasion to occasion as if each sample were being drawn from a distribution. As a consequence, the value of a distractor that is “drawn” during a decision between two other options may be larger than the distractor’s average value and may even have a value that is larger than the value drawn from the less valuable choice option’s distribution on the current trial. On such a trial it may become especially clear that the better of the two options has a higher value than the alternative choice option. Our understanding is that Webb, Louie and colleagues (Louie et al., 2013; Webb et al., 2020) suggest an explanation approximately along these lines when they reported a negative distractor effect during some decisions, i.e., they follow the predictions of divisive normalization suggesting that decisions become more random as the distractor’s value is greater.

An alternative approach, however, assumes that rather than noise in the representation of the option itself, there is noise in the comparison process when the two options are compared. This is exemplified in many influential decision making models including evidence accumulation models such as drift diffusion models (Shadlen & Shohamy, 2016) and recurrent neural network models of decision making (Wang, 2008). It is this latter type of model that we have used in our previous investigations (Chau et al., 2020; Kohl et al., 2023). However, these two approaches are linked both in their theoretical origin and in the predictions that they make in many situations (Shadlen & Shohamy, 2016). We therefore clarify that this is the case in the revised manuscript as follows:

“In the current study and in previous work we have used or made reference to models of decision making that assume that a noisy process of choice comparison occurs such as recurrent neural networks and drift diffusion models (Shadlen & Shohamy, 2016; Wang, 2008). Under this approach, positive distractor effects are predicted when the comparison process becomes more accurate because of an impact on the noisy process of choice comparison (Chau et al., 2020; Kohl et al., 2023). However, it is worth noting that another class of models might assume that a choice representation itself is inherently noisy. According to this approach, on any given decision a sample is drawn from a distribution of value estimates in a noisy representation of the option. Thus, when the representation is accessed, it might have a certain value on average but this value might vary from occasion to occasion. As a consequence, the value of a distractor that is “drawn” during decision between two other options may be larger than the distractor’s average value and may even have a value that is larger than the value drawn from the less valuable choice option’s distribution on the current trial. On such a trial it may become especially clear that the better of the two options has a higher value than the alternative choice option. Louie and colleagues (Louie et al., 2013) suggest an explanation approximately along these lines when they reported a positive distractor effect during some decisions. Such different approaches share theoretical origins (Shadlen & Shohamy, 2016) and make related predictions about the impact of distractors on decision making.” (Lines 297-313)

Reviewer #2 Comment 2(2) How to Interpret the Composite (Mixture) model?On the other side of the correlation are the results from the mixture model for how decision-makers aggregate attributes. The authors report that most subjects are best represented by a mixture of additive and multiplicative aggregation models. The authors justify this with the proposal that these values are computed in different brain regions and then aggregated (which is reasonable, though raises the question of "where" if not the mPFC). However, an equally reasonable interpretation is that the improved fit of the mixture model simply reflects a misspecification of two extreme aggregation processes (additive and EV), so the log-likelihood is maximized at some point in between them.One possibility is a model with utility curvature. How much of this result is just due to curvature in valuation? There are many reasonable theories for why we should expect curvature in utility for human subjects (for example, limited perception: Robson, 2001, Khaw, Li Woodford, 2019; Netzer et al., 2022) and of course many empirical demonstrations of risk aversion for small stakes lotteries. The mixture model, on the other hand, has parametric flexibility.There is also a large literature on testing expected utility jointly with stochastic choice, and the impact of these assumptions on parameter interpretation (Loomes & Sugden, 1998; Apesteguia & Ballester, 2018; Webb, 2019). This relates back to the point above: the mixture may reflect the joint assumption of how choice departs from deterministic EV.

We thank the reviewer for the comment. They are indeed right to mention the vast literature on curvature in subjective valuation; however it is important to stress that the predictions of the additive model with linear basis functions are quite distinct for the predictions of a multiplicative model with non-linear basis functions. We have tested the possibility that participants’ behaviour was better explained by the latter and we showed that this was not the case. Specifically, we have added and performed model fitting on an additional model with utility curvature based on prospect theory (Kahneman & Tversky, 1979) with the weighted probability function suggested by (Prelec, 1998):X¯i=XiαP¯i=e−(−log⁡Pi)βPTi=X¯i×P¯i

where and represent the reward magnitude and probability (both rescaled to the interval between 0 and 1), respectively. is the weighted magnitude and is the weighted probability, while and are the corresponding distortion parameters. This prospect theory (PT) model is included along with the four previous models (please refer to Figure 3) in a Bayesian model comparison. Results indicate that the composite model remains the best account of participants’ choice behaviour (exceedance probability = 1.000, estimated model frequency = 0.720). We have now included these results in the main text and Supplementary Figure 2:

“Supplementary Figure 2 reports an additional Bayesian model comparison performed while including a model with nonlinear utility functions based on Prospect Theory (Kahneman & Tversky, 1979) with the Prelec formula for probability (Prelec, 1998). Consistent with the above finding, the composite model provides the best account of participants’ choice behaviour (exceedance probability = 1.000, estimated model frequency = 0.720).” (Lines 193-198)

Reviewer #2 Comment 31. So then how should we interpret the correlation that the authors report?On one side we have the impact of the binary/ternary treatment which demonstrates some impact of the low value alternative on a binary choice probability. This may reflect some deep flaws in existing theories of choice, or it may simply reflect some departure from purely deterministic expected value maximization that existing theories can address. We have no theory to connect it to, so we cannot tell. On the other side of the correlation, we have a mixture between additive and multiplicative preferences over risk. This result may reflect two distinct neural processes at work, or it may simply reflect a misspecification of the manner in which humans perceive and aggregate attributes of a lottery (or even just the stimuli in this experiment) by these two extreme candidates (additive vs. EV). Again, this would entail some departure from purely deterministic expected value maximization that existing theories can address.It is entirely possible that the authors are reporting a result that points to the more exciting of these two possibilities. But it is also possible (and perhaps more likely) that the correlation is more mundane. The paper does not guide us to theories that predict such a correlation, nor reject any existing ones. In my opinion, we should be striving for theoretically-driven analyses of datasets, where the interpretation of results is clearer.

We thank the reviewer for their clear comments. Based on our responses to the previous comments it should be apparent that our results are consistent with several existing theories of choice, so we are not claiming that there are deep flaws in them, but distinct neural processes (additive and multiplicative) are revealed, and this does not reflect a misspecification in the modelling. We have revised our manuscript in the light of the reviewer’s comments in the hope of clarifying the theoretical background which informed both our data analysis and our data interpretation.

First, we note that there are theoretical reasons to expect a third option might impact on choice valuation. There is a large body of work suggesting that a third option may have an impact on the values of two other options (indeed Reviewer #2 refers to some of this work in their Reviewer #2 Comment 1), but the body of theoretical work originates partly in neuroscience and not just in behavioural economics. In many sensory systems, neural activity changes with the intensity of the stimuli that are sensed. Divisive normalization in sensory systems, however, describes the way in which such neural responses are altered also as a function of other adjacent stimuli (Carandini & Heeger, 2012; Glimcher, 2022; Louie et al., 2011, 2013). The phenomenon has been observed at neural and behavioural levels as a function not just of the physical intensity of the other stimuli but as a function of their associated value (Glimcher, 2014, 2022; Louie et al., 2011, 2015; Noonan et al., 2017; Webb et al., 2020).

Analogously there is an emerging body of work on the combinatorial processes that describe how multiple representational elements are integrated into new representations (Barron et al., 2013; Papageorgiou et al., 2017; Schwartenbeck et al., 2023). These studies have originated in neuroscience, just as was the case with divisive normalization, but they may have implications for understanding behaviour. For example, they might be linked to behavioural observations that the values assigned to bundles of goods are not necessarily the sum of the values of the individual goods (Hsee, 1998; List, 2002). One neuroscience fact that we know about such processes is that, at an anatomical level, they are linked to the medial frontal cortex (Barron et al., 2013; Fellows, 2006; Hunt et al., 2012; Papageorgiou et al., 2017; Schwartenbeck et al., 2023). A second neuroscientific fact that we know about medial frontal cortex is that it is linked to any positive effects that distractors might have on decision making (Chau et al., 2014; Noonan et al., 2017). Therefore, we might make use of these neuroscientific facts and theories to predict a correlation between positive distractor effects and non-additive mechanisms for determining the integrated value of multi-component choices. This is precisely what we did; we predicted the correlation on the basis of this body of work and when we tested to see if it was present, we found that indeed it was. It may be the case that other behavioural economics theories offer little explanation of the associations and correlations that we find. However, we emphasize that this association is predicted by neuroscientific theory and in the revised manuscript we have attempted to clarify this in the Introduction and Discussion sections:

“Given the overlap in neuroanatomical bases underlying the different methods of value estimation and the types of distractor effects, we further explored the relationship. Critically, those who employed a more multiplicative style of integrating choice attributes also showed stronger positive distractor effects, whereas those who employed a more additive style showed negative distractor effects. These findings concur with neural data demonstrating that the medial prefrontal cortex (mPFC) computes the overall values of choices in ways that go beyond simply adding their components together, and is the neural site at which positive distractor effects emerge (Barron et al., 2013; Bongioanni et al., 2021; Chau et al., 2014; Fouragnan et al., 2019; Noonan et al., 2017; Papageorgiou et al., 2017), while divisive normalization was previously identified in the posterior parietal cortex (PPC) (Chau et al., 2014; Louie et al., 2011).” (Lines 109-119)

“At the neuroanatomical level, the negative distractor effect is mediated by the PPC, where signal modulation described by divisive normalization has been previously identified (Chau et al., 2014; Louie et al., 2011). The same region is also crucial for perceptual decision making processes (Shadlen & Shohamy, 2016). The additive heuristics for combining choice attributes are closer to a perceptual evaluation because distances in this subjective value space correspond linearly to differences in physical attributes of the stimuli, whereas normative (multiplicative) value has a non-linear relation with them (cf. Figure 1c). It is well understood that many sensory mechanisms, such as in primates’ visual systems or fruit flies’ olfactory systems, are subject to divisive normalization (Carandini & Heeger, 2012). Hence, the additive heuristics that are more closely based on sensory mechanisms could also be subject to divisive normalization, leading to negative distractor effects in decision making.

In contrast, the positive distractor effect is mediated by the mPFC (Chau et al., 2014; Fouragnan et al., 2019). Interestingly, the same or adjacent, interconnected mPFC regions have also been linked to the mechanisms by which representational elements are integrated into new representations (Barron et al., 2013; Klein-Flügge et al., 2022; Law et al., 2023; Papageorgiou et al., 2017; Schwartenbeck et al., 2023). In a number of situations, such as multi-attribute decision making, understanding social relations, and abstract knowledge, the mPFC achieves this by using a spatial map representation characterised by a grid-like response (Constantinescu et al., 2016; Bongioanni et al., 2021; Park et al., 2021) and disrupting mPFC leads to the evaluation of composite choice options as linear functions of their components (Bongioanni et al., 2021). These observations suggest a potential link between positive distractor effects and mechanisms for evaluating multiple component options and this is consistent with the across-participant correlation that we observed between the strength of the positive distractor effect and the strength of non-additive (i.e., multiplicative) evaluation of the composite stimuli we used in the current task. Hence, one direction for model development may involve incorporating the ideas that people vary in their ways of combining choice attributes and each way is susceptible to different types of distractor effect.” (Lines 250-274)

Reviewer #2 Comment 4(4) Finally, the results from these experiments might not have external validity for two reasons. First, the normative criterion for multi-attribute decision-making differs depending on whether the attributes are lotteries or not (i.e. multiplicative vs additive). Whether it does so for humans is a matter of debate. Therefore if the result is unique to lotteries, it might not be robust for multi-attribute choice more generally. The paper largely glosses over this difference and mixes literature from both domains. Second, the lottery information was presented visually and there is literature suggesting this form of presentation might differ from numerical attributes. Which is more ecologically valid is also a matter of debate.

We thank the reviewer for the comment. Indeed, they are right that the correlation we find between value estimation style and distractor effects may not be detected in all contexts of human behaviour. What the reviewer suggests goes along the same lines as our response to Reviewer #1 Comment 3, multi-attribute value estimation may have different structure: in some cases, the optimal solution may require a non-linear (e.g., multiplicative) response as in probabilistic or delayed decisions, but other cases (e.g., when estimating the value of a snack based on its taste, size, healthiness, price) a linear integration would suffice. In the latter kind of scenarios, both the optimal and the heuristic solutions may be additive and people’s value estimation “style” may not be teased apart. However, if different neural mechanisms associated with difference estimation processes are observed in certain scenarios, it suggests that these mechanisms are always present, even in scenarios where they do not alter the predictions. Probabilistic decision-making is also pervasive in many aspects of daily life and not just limited to the case of lotteries.

While behaviour has been found to differ depending on whether lottery information is presented graphically or numerically, there is insufficient evidence to suggest biases towards additive or multiplicative evaluation, or towards positive or negative distractor effects. As such, we may expect that the correlation that we reveal in this paper, grounded in distinct neural mechanisms, would still hold even under different circumstances.

Taking previous literature as examples, similar patterns of behaviour have been observed in humans when making decisions during trinary choice tasks. In a study conducted by Louie and colleagues (Louie et al., 2013; Webb et al., 2020), human participants performed a snack choice task where their behaviour could be modelled by divisive normalization with biphasic response (i.e., both positive and negative distractor effects). While these two studies only use a single numerical value of price for behavioural modelling, these prices should originate from an internal computation of various attributes related to each snack that are not purely related to lotteries. Expanding towards the social domain, studies of trinary decision making have considered face attractiveness and averageness (Furl, 2016), desirability of hiring (Chang et al., 2019), as well as desirability of candidates during voting (Chang et al., 2019). These choices involve considering various attributes unrelated to lotteries or numbers and yet, still display a combination of positive distractor and negative distractor (i.e. divisive normalization) effects, as in the current study. In particular, the experiments carried out by Chang and colleagues (Chang et al., 2019) involved decisions in a social context that resemble real-world situations. These findings suggests that both types of distractor effects can co-exist in other value based decision making tasks (Li et al., 2018; Louie et al., 2013) as well as decision making tasks in social contexts (Chang et al., 2019; Furl, 2016).

Reviewer #2 Comment 5Minor Issues:The definition of EV as a normative choice baseline is problematic. The analysis requires that EV is the normative choice model (this is why the HV-LV gap is analyzed and the distractor effect defined in relation to it). But if the binary/ternary interaction effect can be accounted for by curvature of a value function, this should also change the definition of which lottery is HV or LV for that subject!

We thank the reviewer for the comment. While the initial part of the paper discussed results that were defined by the EV model, the results shown in Supplementary Figure 2 were generated by replacing the utility function based on values obtained by using the composite model. Here, we have also redefined the definition of HV or LV for each subject depending on the updated value generated by the composite model prior to the regression.

References

Apesteguia, J. & Ballester, M. Monotone stochastic choice models: The case of risk and time preferences. Journal of Political Economy (2018).

Block, H. D. & Marschak, J. Random Orderings and Stochastic Theories of Responses. Cowles Foundation Discussion Papers (1959).

Khaw, M. W., Li, Z. & Woodford, M. Cognitive Imprecision and Small-Stakes Risk Aversion. Rev. Econ. Stud. 88, 1979-2013 (2020).

Loomes, G. & Sugden, R. Testing Different Stochastic Specificationsof Risky Choice. Economica 65, 581-598 (1998).

Luce, R. D. Indvidual Choice Behaviour. (John Wiley and Sons, Inc, 1959).

Netzer, N., Robson, A. J., Steiner, J. & Kocourek, P. Endogenous Risk Attitudes. SSRN Electron. J. (2022) doi:10.2139/ssrn.4024773.

Robson, A. J. Why would nature give individuals utility functions? Journal of Political Economy 109, 900-914 (2001).

Webb, R. The (Neural) Dynamics of Stochastic Choice. Manage Sci 65, 230-255 (2019).

**Reviewer #3 (Public Review):**
Summary:The way an unavailable (distractor) alternative impacts decision quality is of great theoretical importance. Previous work, led by some of the authors of this study, had converged on a nuanced conclusion wherein the distractor can both improve (positive distractor effect) and reduce (negative distractor effect) decision quality, contingent upon the difficulty of the decision problem. In very recent work, Cao and Tsetsos (2022) reanalyzed all relevant previous datasets and showed that once distractor trials are referenced to binary trials (in which the distractor alternative is not shown to participants), distractor effects are absent. Cao and Tsetsos further showed that human participants heavily relied on additive (and not multiplicative) integration of rewards and probabilities.The present study by Wong et al. puts forward a novel thesis according to which interindividual differences in the way of combining reward attributes underlie the absence of detectable distractor effect at the group level. They re-analysed the 144 human participants and classified participants into a "multiplicative integration" group and an "additive integration" group based on a model parameter, the "integration coefficient", that interpolates between the multiplicative utility and the additive utility in a mixture model. They report that participants in the "multiplicative" group show a negative distractor effect while participants in the "additive" group show a positive distractor effect. These findings are extensively discussed in relation to the potential underlying neural mechanisms.Strengths:- The study is forward-looking, integrating previous findings well, and offering a novel proposal on how different integration strategies can lead to different choice biases.- The authors did an excellent job of connecting their thesis with previous neural findings. This is a very encompassing perspective that is likely to motivate new studies towards a better understanding of how humans and other animals integrate information in decisions under risk and uncertainty.- Despite that some aspects of the paper are very technical, methodological details are well explained and the paper is very well written.

We thank the reviewer for the positive response and are pleased that the reviewer found our report interesting.

Reviewer #3 Comment 1Weaknesses:The authors quantify the distractor variable as "DV - HV", i.e., the relative distractor variable. Do the conclusions hold when the distractor is quantified in absolute terms (as "DV", see also Cao & Tsetsos, 2023)? Similarly, the authors show in Suppl. Figure 1 that the inclusion of a HV + LV regressor does not alter their conclusions. However, the (HV + LV)*T regressor was not included in this analysis. Does including this interaction term alter the conclusions considering there is a high correlation between (HV + LV)*T and (DV - HV)*T? More generally, it will be valuable if the authors assess and discuss the robustness of their findings across different ways of quantifying the distractor effect.

We thank the reviewer for the comment. In the original manuscript we had already demonstrated that the distractor effect was related to the integration coefficient using a number of complementary analyses. They include Figure 5 based on GLM2, Supplementary Figure 3 based on GLM3 (i.e., adding the HV+LV term to GLM2), and Supplementary Figure 4 based on GLM2 but applying the utility estimate from the composite model instead of expected value (EV). These three sets of analyses produced comparable results. The reason why we elected not to include the (HV+LV)T term in GLM3 (Supplementary Figure 3) was due to the collinearity between the regressors in the GLM. If this term is included in GLM3, the variance inflation factor (VIF) would exceed an acceptable level of 4 for some regressors. In particular, the VIF for the (HV+LV) and (HV+LV)T regressors is 5.420, while the VIF for (DV−HV) and (DV−HV)T is 4.723.

Here, however, we consider the additional analysis suggested by the reviewer and test whether similar results are obtained. We constructed GLM4 including the (HV+LV)T term but replacing the relative distractor value (DV-HV) with the absolute distractor value (DV) in the main term and its interactions, as follows:

GLM4:=β0+β1z(HV+LV)+β2z(HV−LV)+β3z(DV)+β4z(HV−LV)z(DV)+β5T+β6z(HV+LV)T+β7z(HV−LV)T+β8z(DV)T+β9z(HV−LV)z(DV)T+ε

A significant negative (DV)T effect was found for the additive group [t(72)=−2.0253, p=0.0465] while the multiplicative group had a positive trend despite not reaching significance. Between the two groups, the (DV)T term was significantly different [t(142)=2.0434, p=0.0429]. While these findings suggest that the current conclusions could be partially replicated, simply replacing the relative distractor value with the absolute value in the previous analyses resulted in non-significant findings. Taking these results together with the main findings, it is possible to conclude that the positive distractor effect is better captured using the relative DV-HV term rather than the absolute DV term. This would be consistent with the way in which option values are envisaged to interact with one another in the mutual inhibition model (Chau et al., 2014, 2020) that generates the positive distractor effect. The model suggests that evidence is accumulated as the difference between the excitatory input from the option (e.g. the HV option) and the pooled inhibition contributed partly by the distractor. We have now included these results in the manuscript:

“Finally, we performed three additional analyses that revealed comparable results to those shown in Figure 5. In the first analysis, reported in Supplementary Figure 3, we added an term to the GLM, because this term was included in some analyses of a previous study that used the same dataset (Chau et al., 2020). In the second analysis, we added an term to the GLM. We noticed that this change led to inflation of the collinearity between the regressors and so we also replaced the (DV−HV) term by the DV term to mitigate the collinearity (Supplementary Figure 4). In the third analyses, reported in Supplementary Figure 5, we replaced the utility terms of GLM2. Since the above analyses involved using HV, LV, and DV values defined by the normative Expected Value model, here, we re-defined the values using the composite model prior to applying GLM2. Overall, in the Multiplicative Group a significant positive distractor effect was found in Supplementary Figures 3 and 4. In the Additive Group a significant negative distractor effect was found in Supplementary Figures 3 and 5. Crucially, all three analyses consistently showed that the distractor effects were significantly different between the Multiplicative Group and the Additive Group.” (Lines 225-237)

Reviewer #3 Comment 2The central finding of this study is that participants who integrate reward attributes multiplicatively show a positive distractor effect while participants who integrate additively show a negative distractor effect. This is a very interesting and intriguing observation. However, there is no explanation as to why the integration strategy covaries with the direction of the distractor effect. It is unlikely that the mixture model generates any distractor effect as it combines two "context-independent" models (additive utility and expected value) and is fit to the binary-choice trials. The authors can verify this point by quantifying the distractor effect in the mixture model. If that is the case, it will be important to highlight that the composite model is not explanatory; and defer a mechanistic explanation of this covariation pattern to future studies.

We thank the reviewer for the comment. Indeed, the main purpose of applying the mixture model was to identify the way each participants combined attributes and, as the reviewer pointed out, the mixture model per se is context independent. While we acknowledge that the mixture model is not a mechanistic explanation, there is a theoretical basis for the observation that these two factors are linked.

Firstly, studies that have examined the processes involved when humans combine and integrate different elements to form new representations (Barron et al., 2013; Papageorgiou et al., 2017; Schwartenbeck et al., 2023) have implicated the medial frontal cortex as a crucial region (Barron et al., 2013; Fellows, 2006; Hunt et al., 2012; Papageorgiou et al., 2017; Schwartenbeck et al., 2023). Meanwhile, previous studies have also identified that positive distractor effects are linked to the medial frontal cortex (Chau et al., 2014; Noonan et al., 2017). Therefore, the current study utilized these two facts to establish the basis for a correlation between positive distractor effects and non-additive mechanisms for determining the integrated value of multi-component choices. Nevertheless, we agree with the reviewer that it will be an important future direction to look at how the covariation pattern emerges in a computational model. We have revised the manuscript in an attempt to address this issue.

“At the neuroanatomical level, the negative distractor effect is mediated by the PPC, where signal modulation described by divisive normalization has been previously identified (Chau et al., 2014; Louie et al., 2011). The same region is also crucial for perceptual decision making processes (Shadlen & Shohamy, 2016). The additive heuristics for combining choice attributes are closer to a perceptual evaluation because distances in this subjective value space correspond linearly to differences in physical attributes of the stimuli, whereas normative (multiplicative) value has a non-linear relation with them (cf. Figure 1c). It is well understood that many sensory mechanisms, such as in primates’ visual systems or fruit flies’ olfactory systems, are subject to divisive normalization (Carandini & Heeger, 2012). Hence, the additive heuristics that are more closely based on sensory mechanisms could also be subject to divisive normalization, leading to negative distractor effects in decision making.

In contrast, the positive distractor effect is mediated by the mPFC (Chau et al., 2014; Fouragnan et al., 2019). Interestingly, the same or adjacent, interconnected mPFC regions have also been linked to the mechanisms by which representational elements are integrated into new representations (Barron et al., 2013; Klein-Flügge et al., 2022; Law et al., 2023; Papageorgiou et al., 2017; Schwartenbeck et al., 2023). In a number of situations, such as multi-attribute decision making, understanding social relations, and abstract knowledge, the mPFC achieves this by using a spatial map representation characterised by a grid-like response (Constantinescu et al., 2016; Bongioanni et al., 2021; Park et al., 2021) and disrupting mPFC leads to the evaluation of composite choice options as linear functions of their components (Bongioanni et al., 2021). These observations suggest a potential link between positive distractor effects and mechanisms for evaluating multiple component options and this is consistent with the across-participant correlation that we observed between the strength of the positive distractor effect and the strength of non-additive (i.e., multiplicative) evaluation of the composite stimuli we used in the current task. Hence, one direction for model development may involve incorporating the ideas that people vary in their ways of combining choice attributes and each way is susceptible to different types of distractor effect.” (Lines 250-274)

Reviewer #3 Comment 3- Correction for multiple comparisons (e.g., Bonferroni-Holm) was not applied to the regression results. Is the "negative distractor effect in the Additive Group" (Fig. 5c) still significant after such correction? Although this does not affect the stark difference between the distractor effects in the two groups (Fig. 5a), the classification of the distractor effect in each group is important (i.e., should future modelling work try to capture both a negative and a positive effect in the two integration groups? Or just a null and a positive effect?).

We thank the reviewer for the comment. We have performed Bonferroni-Holm correction and as the reviewer surmised, the negative distractor effect in the additive group becomes non-significant. However, we have to emphasize that our major claim is that there was a covariation between decision strategy (of combining attributes) and distractor effect (as seen in Figure 4). That analysis does not imply multiple comparisons. The analysis in Figure 5 that splits participants into two groups was mainly designed to illustrate the effects for an easier understanding by a more general audience. In many cases, the precise ways in which participants are divided into subgroups can have a major impact on whether each individual group’s effects are significant or not. It may be possible to identify an optimal way of grouping, but we refrained from taking such a trial-and-error approach, especially for the analysis in Figure 5 that simply supplements the point made in Figure 4. The key notion we would like the readers to take away is that there is a spectrum of distractor effects (ranging from negative to positive) that will vary depending on how the choice attributes were integrated.

**Reviewer #1 (Recommendations For The Authors):**
Reviewer #1 Recommendations 1Enhancements are necessary for the quality of the scientific writing. Several sentences have been written in a negligent manner and warrant revision to ensure a higher level of rigor. Moreover, a number of sentences lack appropriate citations, including but not restricted to:- Line 39-41.- Line 349-350 (also please clarify what it means by parameter estimate" is very accurate: correlation?).

We thank the reviewer for the comment. We have made revisions to various parts of the manuscript to address the reviewer’s concerns.

“Intriguingly, most investigations have considered the interaction between distractors and chooseable options either at the level of their overall utility or at the level of their component attributes, but not both (Chau et al., 2014, 2020; Gluth et al., 2018).” (Lines 40-42)

“Additional simulations have shown that the fitted parameters can be recovered with high accuracy (i.e., with a high correlation between generative and recovered parameters).” (Lines 414-416)

Reviewer #1 Recommendations 2Some other minor suggestions:- Correlative vs. Causality: the manuscript exhibits a lack of attentiveness in drawing causal conclusions from correlative evidence (manuscript title, Line 91, Line 153-155).- When displaying effect size on accuracy, there is no need to show the significance of intercept (Figure 2,5, & supplementary figures).- Adding some figure titles on Figure 2 so it is clear what each panel stands for.- In Figure 3, the dots falling on zero values are not easily seen. Maybe increasing the dot size a little?- Line 298: binomial linking function (instead of binomial distribution).- Line 100: composite, not compositive.- Line 138-139: please improve the sentence, if it's consistent with previous findings, what's the point of "surprisingly"?

We thank the reviewer for the suggestions. We have made revisions to the title and various parts of the manuscript to address the reviewer’s concerns.

- Correlative vs. Causality: the manuscript exhibits a lack of attentiveness in drawing causal conclusions from correlative evidence (manuscript title, Line 91, Line 153-155).

We have now revised the manuscript:

“Distractor effects in decision making are related to the individual’s style of integrating choice attributes” (title of the manuscript)

“More particularly, we consider whether individual differences in combination styles could be related to different forms of distractor effect.” (Lines 99-100)

“While these results may seem to suggest that a distractor effect was not present at an overall group level, we argue that the precise way in which a distractor affects decision making is related to how individuals integrate the attributes.” (Lines 164-167)

- When displaying effect size on accuracy, there is no need to show the significance of intercept (Figure 2,5, & supplementary figures).

We have also modified all Figures to remove the intercept.

- Adding some figure titles on Figure 2 so it is clear what each panel stands for.

We have added titles accordingly.

- In Figure 3, the dots falling on zero values are not easily seen. Maybe increasing the dot size a little?

In conjunction with addressing Reviewer #3 Recommendation 6, we have adapted the violin plots into histograms for a better representation of the values.

- Line 298: binomial linking function (instead of binomial distribution).

- Line 100: composite, not compositive.

- Line 138-139: please improve the sentence, if it's consistent with previous findings, what's the point of "surprisingly"?

We have made revisions accordingly.

**Reviewer #2 (Recommendations For The Authors):**
Reviewer #2 Recommendations 1Line 294. The definition of DV, HV, LV is not sufficient. Presumably, these are the U from the following sections? Or just EV? But this is not explicitly stated, rather they are vaguely referred to as values." The computational modelling section refers to them as utilities. Are these the same thing?

We thank the reviewer for the suggestion. We have clarified that the exact method for calculating each of the values and updated the section accordingly.

“where HV, LV, and DV refer to the values of the chooseable higher value option, chooseable lower value option, and distractor, respectively. Here, values (except those in Supplementary Figure 5) are defined as Expected Value (EV), calculated by multiplying magnitude and probability of reward.” (Lines 348-350)

Reviewer #2 Recommendations 2The analysis drops trials in which the distractor was chosen. These trials are informative about the presence (or not) of relative valuation or other factors because they make such choices more (or less) likely. Ignoring them is another example of the analysis being misspecified.

We thank the reviewer for the suggestion and this is related to Major Issue 1 raised by the same reviewer. In brief, we adopted the same methods implemented by Cao and Tsetsos (Cao and Tsetsos, 2022) and that constrained us to applying a binomial model. Please refer to our reply to Major Issue 1 for more details.

Reviewer #2 Recommendations 3Some questions and suggestions on statistics and computational modeling:Have the authors looked at potential collinearity between the regressors in each of the GLMs?

We thank the reviewer for the comment. For each of the following GLMs, the average variance inflation factor (VIF) has been calculated as follows:

GLM2 using the Expected Value model: Logit(accuracy) =β0+β1z(HV−LV)+β2z(DV−HV)+β3z(HV−LV)z(DV−HV)+β4T+β5z(HV−LV)T+β6z(DV−HV)T+β7z(HV−LV)z(DV−HV)T+ε

**Author response table 1. sa4table1:** 

Regressor	beta_(1)	beta_(2)	beta_(3)	beta_(4)	beta_(5)	beta_(6)	beta_(7)
VIF	2.0799	2.0885	2.0089	1	2.0799	2.0885	2.0089

GLM2 after replacing the utility function based on the normative Expected Value model with values obtained by using the composite model: Logit(accuracy) =β0+β1z(HV−LV)+β2z(DV−HV)+β3z(HV−LV)z(DV−HV)+β4T+β5z(HV−LV)T+β6z(DV−HV)T+β7z(HV−LV)z(DV−HV)T+ε

**Author response table 2. sa4table2:** 

Regressor	beta_(1)	beta_(2)	beta_(3)	beta_(4)	beta_(5)	beta_(6)	beta_(7)
VIF	2.0600	2.1433	2.1360	1	2.0600	2.1433	2.1360

GLM3: Logit(accuracy) =β0+β1z(HV+LV)+β2z(HV−LV)+β3z(DV−HV)+β4z(HV−LV)z(DV−HV)+β5T+β6z(HV−LV)T+β7z(DV−HV)T+β8z(HV−LV)z(DV−HV)T+ε

**Author response table 3. sa4table3:** 

Regressor	beta_(1)	beta_(2)	beta_(3)	beta_(4)	beta_(5)	beta_(6)	beta_(7)	beta_(8)
VIF	2.5814	2.1014	3.7051	2.0136	1	2.0799	2.0885	2.0089

As indicated in the average VIF values calculated, none of them exceed 4, suggesting that the estimated coefficients were not inflated due to collinearity between the regressor in each of the GLMs.

Reviewer #2 Recommendations 4- Correlation results in Figure 4. What is the regression line displayed on this plot? I suspect the regression line came from Pearson's correlation, which would be inconsistent with the Spearman's correlation reported in the text. A reasonable way would be to transform both x and y axes to the ranked data. However, I wonder why it makes sense to use ranked data for testing the correlation in this case. Those are both scalar values. Also, did the authors assess the influence of the zero integration coefficient on the correlation result? Importantly, did the authors redo the correlation plot after defining the utility function by the composite models?

We thank the reviewer for the suggestion. The plotted line in Figure 4 was based on the Pearson’s correlation and we have modified the text to also report the Pearson’s correlation result as well.

If we were to exclude the 32 participants with integration coefficients smaller than 1×10-6 from the analysis, we still observe a significant positive Pearson’s correlation [r(110)=0.202, p=0.0330].

**Author response image 1. sa4fig1:** Figure 4 after excluding 32 participants with integration coefficients smaller than 1×10-6.

“As such, we proceeded to explore how the distractor effect (i.e., the effect of (DV−HV)T obtained from GLM2; Figure 2c) was related to the integration coefficient (η) of the optimal model via a Pearson’s correlation (Figure 4). As expected, a significant positive correlation was observed [r(142)=0.282, p=0.000631]. We noticed that there were 32 participants with integration coefficients that were close to zero (below 1×10-6). The correlation remained significant even after removing these participants [r(110)=0.202, p=0.0330].” (Lines 207-212)

The last question relates to results already included in Supplementary Figure 5, in which the analyses were conducted using the utility function of the composite model. We notice that although there was a difference in integration coefficient between the multiplicative and additive groups, a correlational analysis did not generate significant results [r(142)=0.124, p=0.138]. It is possible that the relationship became less linear after applying the composite model utility function. However, it is noticeable that in a series of complementary analyses (Figure 5: r(142)=0.282, p=0.000631; Supplementary Figure 3: r(142)=0.278, p=0.000746) comparable results were obtained.

Reviewer #2 Recommendations 5- From lines 163-165, were the models tested on only the three-option trials or both two and three-opinion trials? It is ambiguous from the description here. It might be worth checking the model comparison based on different trial types, and the current model fitting results do not tell an absolute sense of the goodness of fit. I would suggest including the correctly predicted trial proportions in each trial type from different models.

We thank the reviewer for the suggestion. We have only modeled the two-option trials and the key reason for this is because the two-option trials can arguably provide a better estimate of participants’ style of integrating attributes as they are independent of any distractor effects. This was also the same reason why Cao and Tsetsos applied the same approach when they were re-analyzing our data (Cao and Tsetsos, 2022). We have clarified the statement accordingly.

“We fitted these models exclusively to the Two-Option Trial data and not the Distractor Trial data, such that the fitting (especially that of the integration coefficient) was independent of any distractor effects, and tested which model best describes participants’ choice behaviours.” (Lines 175-178)

Reviewer #2 Recommendations 6- Along with displaying the marginal distributions of each parameter estimate, a correlation plot of these model parameters might be useful, given that some model parameters are multiplied in the value functions.

We thank the reviewer for the suggestion. We have also generated the correlation plot of the model parameters. The Pearson’s correlation between the magnitude/probability weighting and integration coefficient was significant [r(142)=−0.259, p=0.00170]. The Pearson’s correlation between the inverse temperature and integration coefficient was not significant [r(142)=−0.0301, p=0.721]. The Pearson’s correlation between the inverse temperature and magnitude/probability weighting was not significant [r(142)=−0.0715, p=0.394].

“Our finding that the average integration coefficient was 0.325 coincides with previous evidence that people were biased towards using an additive, rather than a multiplicative rule. However, it also shows rather than being fully additive (=0) or multiplicative (=1), people’s choice behaviour is best described as a mixture of both. Supplementary Figure 1 shows the relationships between all the fitted parameters.” (Lines 189-193)

Reviewer #2 Recommendations 7Have the authors tried any functional transformations on amounts or probabilities before applying the weighted sum? The two attributes are on entirely different scales and thus may not be directly summed together.

We thank the reviewer for the comment. Amounts and probabilities were indeed both rescaled to the 0-1 interval before being summed, as explained in the methods (Line XXX). Additionally, we have now added and performed model fitting on an additional model with utility curvature based on the prospect theory (Kahneman & Tversky, 1979) and a weighted probability function (Prelec, 1998):X¯i=XiαP¯i=e−(−log⁡Pi)βPTi=X¯i×P¯i

where and represent the reward magnitude and probability (both rescaled to the interval between 0 and 1), respectively. is the weighted magnitude and is the weighted probability, while and are the corresponding distortion parameters. This prospect theory (PT) model was included along with the four previous models (please refer to Figure 3) in a Bayesian model comparison. Results indicate that the composite model remains as the best account of participants’ choice behaviour (exceedance probability = 1.000, estimated model frequency = 0.720).

“Supplementary Figure 2 reports an additional Bayesian model comparison performed while including a model with nonlinear utility functions based on Prospect Theory (Kahneman & Tversky, 1979) with the Prelec formula for probability (Prelec, 1998). Consistent with the above finding, the composite model provides the best account of participants’ choice behaviour (exceedance probability = 1.000, estimated model frequency = 0.720).” (Lines 193-198)

**Reviewer #3 (Recommendations For The Authors):**
Reviewer #3 Recommendations 1- In the Introduction (around line 48), the authors make the case that distractor effects can co-exist in different parts of the decision space, citing Chau et al. (2020). However, if the distractor effect is calculated relative to the binary baseline this is no longer the case.- Relating to the above point, it might be useful for the authors to make a distinction between effects being non-monotonic across the decision space (within individuals) and effects varying across individuals due to different strategies adopted. These two scenarios are conceptually distinct.

We thank the reviewer for the comment. Indeed, the ideas that distractor effects may vary across decision space and across different individuals are slightly different concepts. We have now revised the manuscript to clarify this:

“However, as has been argued in other contexts, just because one type of distractor effect is present does not preclude another type from existing (Chau et al., 2020; Kohl et al., 2023). Each type of distractor effect can dominate depending on the dynamics between the distractor and the chooseable options. Moreover, the fact that people have diverse ways of making decisions is often overlooked. Therefore, not only may the type of distractor effect that predominates vary as a function of the relative position of the options in the decision space, but also as a function of each individual’s style of decision making.” (Lines 48-54)

Reviewer #3 Recommendations 2- The idea of mixture models/strategies has strong backing from other Cognitive Science domains and will appeal to most readers. It would be very valuable if the authors could further discuss the potential level at which their composite model might operate. Are the additive and EV quantities computed and weighted (as per the integration coefficient) within a trial giving rise to a composite decision variable? Or does the integration coefficient reflect a probabilistic (perhaps competitive) selection of one strategy on a given trial? Perhaps extant neural data can shed light on this question.

We thank the reviewer for the comment. The idea is related to whether the observed mixture in integration models derives from value being actually computed in a mixed way within each trial, or each trial involves a probabilistic selection between the additive and multiplicative strategies. We agree that this is an interesting question and to address it would require the use of some independent continuous measures to estimate the subjective values in quantitative terms (instead of using the categorical choice data). This could be done by collecting pupil size data or functional magnetic resonance imaging data, as the reviewer has pointed out. Although the empirical work is beyond the scope of the current behavioural study, it is worth bringing up this point in the Discussion:

“The current finding involves the use of a composite model that arbitrates between the additive and multiplicative strategies. A general question for such composite models is whether people mix two strategies in a consistent manner on every trial or whether there is some form of probabilistic selection occurring between the two strategies on each trial such that only one strategy is used on any given trial while, on average, one strategy is more probable than the other. To test which is the case requires an independent estimation of subjective values in quantitative terms, such as by pupillometry or functional neuroimaging. Further understanding of this problem will also provide important insight into the precise way in which distractor effects operate at the single-trial level.” (Lines 275-282)

Reviewer #3 Recommendations 3Line 80 "compare pairs of attributes separately, without integration". This additive rule (or the within-attribute comparison) implies integration, it is just not multiplicative integration.

We thank the reviewer for the comment. We have made adjustments to the manuscript to ensure that the message delivered within this manuscript is consistent.

“For clarity, we stress that the same mathematical formula for additive value can be interpreted as meaning that (1) subjects first estimate the value of each option in an additive way (value integration) and then compare the options, or (2) subjects compare the two magnitudes and separately compare the two probabilities without integrating dimensions into overall values. On the other hand, the mathematical formula for multiplicative value is only compatible with the first interpretation. In this paper we focus on attribute combination styles (multiplicative vs additive) and do not make claims on the order of the operations. More particularly, we consider whether individual differences in combination styles could be related to different forms of distractor effect.” (Lines 92-100)

Reviewer #3 Recommendations 4- Not clear why the header in line 122 is phrased as a question.

We thank the reviewer for the suggestion. We have modified the header to the following:

“The distractor effect was absent on average” (Line 129)

Reviewer #3 Recommendations 5- The discussion and integration of key neural findings with the current thesis are outstanding. It might help the readers if certain statements such as "the distractor effect is mediated by the PPC" (line 229) were further unpacked.

We thank the reviewer for the suggestion. We have made modifications to the original passage to further elaborate the statement.

“At the neuroanatomical level, the negative distractor effect is mediated by the PPC, where signal modulation described by divisive normalization has been previously identified (Chau et al., 2014; Louie et al., 2011). The same region is also crucial for perceptual decision making processes (Shadlen & Shohamy, 2016).” (Lines 250-253)

Reviewer #3 Recommendations 6- In Fig. 3c, there seem to be many participants having the integration coefficient close to 0 but the present violin plot doesn't seem to best reflect this highly skewed distribution. A histogram would be perhaps better here.

We thank the reviewer for the suggestion. We have modified the descriptive plots to use histograms instead of violin plots.

“Figures 3c, d and e show the fitted parameters of the composite model: , the integration coefficient determining the relative weighting of the additive and multiplicative value ( , ); , the magnitude/probability weighing ratio ( , ); and , the inverse temperature ( , ). Our finding that the average integration coefficient was 0.325 coincides with previous evidence that people were biased towards using an additive, rather than a multiplicative rule.” (Lines 186-191)